# SpaEF: Spatially Resolved Transcriptomics Data Element-Wise Denoising Framework Powered by Large Models

**Zekuan Shang** [1 2]  **Xiaosong Han** [1]  **Liupu Wang** [1]  **Wei Du** [1]  **Peng Zhao** [1]  **Yuanshu Li** [1]  **Yubin Xiao** [1]  **Xuan Wu** [1]  **You Zhou** [3]

## Abstract

For denoising Spatially Resolved Transcriptomics (SRT) data, existing methods often construct spot and gene graphs to model inter-spot and inter-gene relationships, respectively. However, these methods often introduce spurious similarity biases among spots when constructing the spot graph and fail to capture nonlinear relationships among genes when constructing the gene graph. Moreover, ineffective graph fusion strategies further bottleneck denoising performance. To address these challenges, we propose SpaEF, which innovatively constructs spot and gene graphs with two Large Models (LMs) to inject prior knowledge for mitigating biases and capture nonlinear relationships, and then fuses them with the proposed element-wise graph autoencoder. As far as we know, **SpaEF is the first SRT denoising method that utilizes pre-trained LMs to construct spot and gene graphs.** Experiments on four real-world datasets with corresponding downstream tasks demonstrate that SpaEF not only outperforms SOTA denoising methods in accuracy but also exhibits strong robustness across tasks.

## 1. Introduction

Spatially Resolved Transcriptomics (SRT) is a cutting-edge technique that provides biologists with rich insight into single-cell biology (Zhao et al., 2025; Oh et al., 2025). However, SRT data can be affected by biotechnological limita-

---

[1]College of Computer Science and Technology, Jilin University, Changchun, China [2]Key Laboratory of Symbolic Computation and Knowledge Engineering of Ministry of Education, Changchun, China [3]College of Software, Jilin University, Changchun, China. Correspondence to: Yubin Xiao <xiaoyb21@mails.jlu.edu.cn>, Xuan Wu <wuuu22@mails.jlu.edu.cn>, You Zhou <zyou@jlu.edu.cn>.

*Proceedings of the 43rd International Conference on Machine Learning*, Seoul, South Korea. PMLR 306, 2026. Copyright 2026 by the author(s).

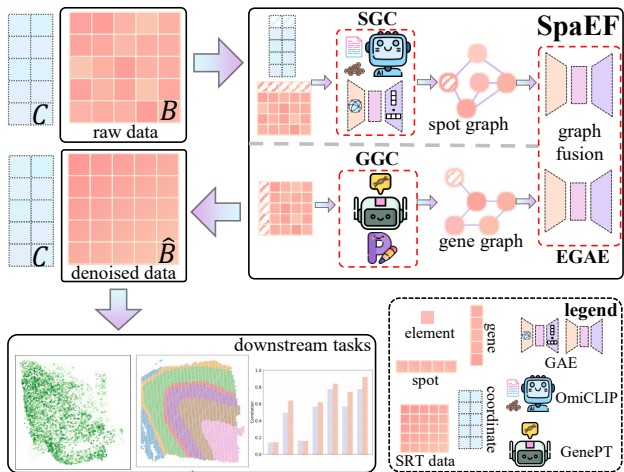

*Figure 1.* Illustration of the SpaEF denoising process. SRT data consists of an expression matrix and spot coordinates that indicate the relative spatial position of each spot. SpaEF leverages pre-trained LMs to capture spot and gene features, respectively, and then integrates these features via the EGAE module, thereby performing element-wise denoising.

tions (e.g., liquid phase diffusion) and operator-dependent variability, introducing significant noise that compromises downstream analyses (Sun et al., 2025a). Thus, denoising SRT data is critical for reliably analyzing spatial gene expression patterns in downstream analysis (Zhu et al., 2025b).

Existing studies primarily employ graph-regularized methods or Graph Neural Networks (GNNs) to denoise SRT data. Graph-regularized methods construct the relationship among spots based on spatial coordinates $C$ and apply graph regularization to smooth the expression matrix $B$, encouraging connected spots to exhibit similar projections in low-dimensional space and thereby reducing noise (Yang et al., 2024; Su et al., 2023). Unlike graph-regularized methods, prior GNN-based approaches generally model spot similarity using expression matrix and spot coordinates, while certain methods further augment the spot graph with histological image-derived features (Hu et al., 2021). But due to the narrow nature and staining-dependent variability of the histological images, incorporating histological images can amplify spurious similarity biases among spots (Kang et al., 2025; Lin et al., 2024; Chen et al., 2020). Moreover,

certain datasets lack histology images, making reliance on them impractical (Wang et al., 2022). Despite this, directly discarding histological images would forfeit structural and morphological features that are often correlated with gene expression and that can provide useful priors for denoising (Pham et al., 2023). Hence, it is important to investigate alternative feature sources that retain structural and morphological information while mitigating spurious biases.

In addition, these earlier studies exclusively focus on spot graph construction and overlook critical gene relationships, which limits denoising performance (Zuo et al., 2024). Recently, certain studies construct gene graphs to model gene relationships and subsequently fuse spot and gene features to produce the final denoised expression matrix. (Madhu et al., 2025). For example, Sun et al. (2025b) proposed DeepGFT, which constructs the gene graph based on statistical co-expression (i.e., Pearson correlation) among genes and then fuses spot and gene graphs via matrix-wise weighting. However, gene relationships extend beyond simple co-expression, and often involve nonlinear mechanisms like path sharing and structural domain sharing (Yuan & Bar-Joseph, 2020), rendering co-expression analysis alone insufficient for gene graph construction. Moreover, during graph fusion, using matrix-wise weighting enforces a single global weight across all entries of the expression matrix, which limits the model's ability to adaptively denoise individual elements and thus potentially degrade denoising performance. Hence, it is valuable to incorporate nonlinear relationships into the gene graph and assign element-wise weights to the fused matrices.

To address these three challenges, we propose the Large Models (LMs)-powered **Spa**tially Resolved Transcriptomics data **E**lement-wise Denoising **F**ramework, named **SpaEF**, which comprises three modules, namely Spot Graph Construction (SGC), Gene Graph Construction (GGC), and Element-wise Graph Autoencoder (EGAE), as shown in Figure 1. Specifically, rather than relying on narrow histological images, SGC employs the text encoder of OmiCLIP (Chen et al., 2025) as a frozen spot encoder to extract spot features. Notably, OmiCLIP employs Contrastive Learning to jointly train its text and image encoders on 2.2 million spot sentence-image pairs (see section 3.1 for more details about spot sentence), thereby learning a shared cross-modal embedding space (Wang et al., 2024a; Zhang et al., 2022). Consequently, by leveraging knowledge acquired during large-scale pre-training, the text encoder can serve as an alternative feature source and produce features with weak biases for spatial spots, producing features that already encode structural and expression-related information. To effectively integrate these OmiCLIP-derived semantic features with spatial information of spots, we further propose a Graph Attention (GAT)-based GAE (AGAE), thereby constructing the spot graph $G^s = (\boldsymbol{B}^T, \mathcal{N}^s)$. By incorporating the GAT

mechanism, AGAE preserves distinctions among neighboring spot embeddings and mitigates over-smoothing. Meanwhile, GGC leverages GenePT, another LM pre-trained on large-scale corpora of gene function descriptions, to capture complex, nonlinear gene relationships (e.g., path sharing) that complement the co-expression relationship, thereby constructing the gene graph $G^g = (\boldsymbol{B}, \mathcal{N}^g)$. Besides, GenePT provides a logistic regression predictor that enables direct inference of the interactions among genes without additional downstream modeling. **As far as we know, SpaEF is the first SRT data denoising work that leverages pre-trained LMs to provide external biological and semantic priors, thereby constructing spot and gene graphs.** Importantly, by leveraging these two robust pre-trained LMs, SpaEF achieves high-level performance across diverse downstream tasks without requiring additional fine-tuning of these two LMs. In contrast, existing LM-based methods (e.g., scGPT) primarily employ LMs as representation learners to encode input data, which may produce low-quality embeddings in zero-shot settings (Kedzierska et al., 2025) (see Section 4 and Appendix F). Moreover, these methods typically require additional fine-tuning, which is both time-consuming and computationally expensive. Therefore, although these approaches also utilize pre-trained LMs, their usage paradigm differs fundamentally from that of SpaEF. Finally, EGAE employs two Feature Autoencoders (FAEs) to independently reconstruct spot and gene graphs, and then performs Element-wise Weighting Addition (EWA) to fuse the reconstructed spot matrix $\boldsymbol{B}^s$ and gene matrix $\boldsymbol{B}^g$, yielding the final denoised matrix $\hat{\boldsymbol{B}}$. Compared to matrix-wise weighting, EGAE assigns a weight to each element, preserving element heterogeneity and enabling the fine-grained joint fusion of spot and gene features.

The key contributions of this work are as follows:

**I)** We incorporate OmiCLIP, a pre-trained multi-modal LM, to extract spot features that capture both structure and expression-related information, mitigating spurious similarity biases that arise from extracting histology features.

**II)** To effectively integrate the spatial information with OmiCLIP-derived semantic features, we propose AGAE, which mitigates over-smoothing among neighboring spots.

**III)** We incorporate GenePT, an LM pre-trained on corpora of gene descriptions, to extract nonlinear relationships among genes beyond the co-expression relationship.

**IV)** We propose EGAE, which performs element-wise addition to enable effective message passing between spot and gene graphs, thereby improving denoising performance.

**V)** To evaluate the performance of the proposed SpaEF, we conduct extensive experiments on four real-world SRT datasets. The experimental results demonstrate that SpaEF

outperforms the SOTA denoising method in terms of accuracy and robustness across different downstream tasks. Moreover, ablation studies and structural analyses demonstrate that the effectiveness of SpaEF stems from the synergy between the LM-based spot and gene graph construction modules and the AGAE and EGAE architectures.

## 2. Related work

In this section, we review the relevant literature.

**Spot Graph Construction Methods:** Due to substantial noise in SRT data, which impairs downstream analyses, prior studies have developed numerous denoising methods. The pioneering denoising methods primarily employ spot-wise techniques (Tang et al., 2023; Yang et al., 2024; Lv et al., 2024). Specifically, these methods employ GNNs to model each spot as a graph node by integrating expression matrices with histological images and/or coordinates, and then define edges according to the similarity of nodes, thereby constructing the spot graph (Hu et al., 2021; Wang et al., 2024b). For example, Tang et al. (2023) combined histological images and expression matrix to construct the spot graph. However, these histological-based methods may amplify spurious similarity biases among spots due to the narrow nature of histological images (Kang et al., 2025). To better address this challenge, SpaEF adopts OmiCLIP, an LM pretrained on large-scale spot expression-image pairs, to extract spot features that encode structural and expression-related information. This design obviates the need for histological input, thereby reducing the risk of introducing spurious similarities. Accordingly, we propose the AGAE module to effectively integrate these OmiCLIP-derived features with the spatial feature of spots (see Section 3.1).

**Gene Graph Construction Methods:** To achieve superior denoising performance in SRT data, recent studies have emphasized the importance of element-wise message passing by modeling gene relationships alongside spot features. Specifically, to encode gene relationships, Zhu et al. (2025a) proposed DUSTED, which exploits gene channel attention to distinguish noisy levels among genes. Madhu et al. (2025) constructed the gene regulatory network within each spot of the spot graph. Sun et al. (2025b) proposed DeepGFT, which explicitly models gene relationships via co-expression analysis. However, DeepGFT primarily relies on analyzing co-expression and may overlook complex, nonlinear gene relationships that commonly occur in real biological systems, resulting in a homogenization of genes (Holdener & De Vlaminck, 2025). In fact, LMs pretrained on large-scale corpora of gene function descriptions can impart rich, nonlinear biological semantics to gene embeddings (Li et al., 2025b). Hence, we adopt GenePT (Chen & Zou, 2025) to generate high-dimensional features for each gene that capture complex, nonlinear relationships, e.g.,

pathways and structural domains, rather than relying solely on co-expression analysis to construct gene graphs. Furthermore, unlike other LMs (e.g., scGPT (Cui et al., 2024) and CellFM (Zeng et al., 2025)) that only produce embeddings, GenePT also provides a logistic regression predictor, enabling direct inference of the interactions among genes without additional downstream modeling.

**Feature Fusion Strategies:** For these element-wise methods, another primary challenge is how to effectively fuse the encoded spot and gene graphs to produce the final denoised expression matrix. DeepGFT (Sun et al., 2025b) explicitly fused spot- and gene-wise denoised matrices via matrix-wise weighting. However, this strategy assigns a uniform weight to all elements of the expression matrix, hindering element-wise denoising and degrading overall performance. To better address this limitation, we propose the EGAE module, which effectively fuses the spot graph and gene graph through element-wise fusion.

## 3. SpaEF

The framework of SpaEF is illustrated in Figure 2. Let $\boldsymbol{B} \in \mathbb{R}^{M \times N}$ denote the expression matrix and $\boldsymbol{C} \in \mathbb{R}^{M \times 2}$ the corresponding spot coordinates $(x, y)$, where $M$ and $N$ denote the number of spots and genes, respectively. The objective of SpaEF is to produce a denoised expression matrix $\hat{\boldsymbol{B}} \in \mathbb{R}^{M \times N}$. This section presents three key modules of SpaEF, namely SGC, GGC, and EGAE. The training procedure of SpaEF is shown in Appendix A.

### 3.1. Spot Graph Construction (SGC) Module

Following the prior study SEDR (Xu et al., 2024), during the construction of the spot graph $G^s$, SGC first constructs a spatial graph $G^a$ to fully use the spot coordinates for spatially local feature extraction. Let $G^a = (\boldsymbol{B}^a, \mathcal{N}^a)$, where $\boldsymbol{b}_i^a$ and $\mathcal{N}_i^a$ denote the feature vector of the $i$th spot and its neighbor set, respectively. To extract spot feature $\boldsymbol{b}_i^a$, prior studies directly exploit encoders to extract narrow image features, resulting in spurious similarity (Hu et al., 2021; Kang et al., 2025). Additionally, certain datasets lack histology images, making reliance on them impractical (Wang et al., 2022). To better address this problem, SGC uniquely incorporates the text encoder from OmiCLIP (Chen et al., 2025) into the SRT data denoising task. OmiCLIP adopts Contrastive Learning to jointly train its text and image encoders on 2.2 million spot sentence-image pairs to learn a shared cross-modal embedding space (Chen et al., 2025; Zhang et al., 2022). Here, a spot sentence is a text string formed by concatenating the names of elements within a spot. After Contrastive Learning, the text encoder can extract both structural and expression-related information directly from spot sentences, obviating the need for incorporating histological images. This design enables SpaEF to achieve high-level

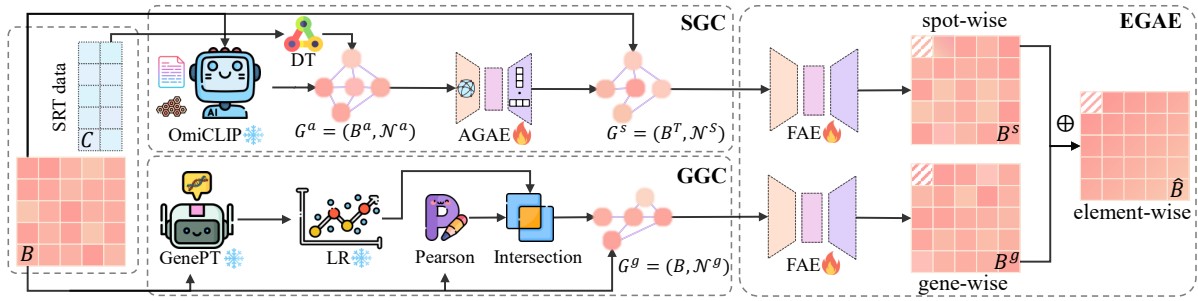

*Figure 2.* Overview of the proposed SpaEF. SGC employs the OmiCLIP text encoder and DT to construct a temporary spatial graph, which AGAE then processes to form the spot graph. GGC employs GenePT to capture gene relationships that complement co-expression correlation for constructing the gene graph. Finally, EGAE utilizes two FAEs to denoise the spot and gene graphs, respectively, and then performs EWA to fuse spot- and gene-wise denoised matrices, yielding the final denoised SRT data.

performance across diverse datasets (see Section 4).

Specifically, SGC first concatenates the names of the ranked $k_e$ most highly expressed non-housekeeping elements in each spot into a spot sentence, thereby capturing the expression pattern of the spot without requiring all element names. Then, the OmiCLIP text encoder processes this sentence to produce the semantic embedding $\boldsymbol{b}_i^a \in \mathbb{R}^{d_0}$ for the $i$th spot ($i \in \{1, 2, \ldots, M\}$), where $d_0$ denotes the embedding dimension. Let $\mathcal{P}_i^a$ denote the set of the top-$k_e$ expressed elements for the spot $i$, $V_j$ the name of the element $j$. The text encoder formally encodes each spot as follows:

$$\boldsymbol{b}_i^a = \text{TextEnc}(\|_{j \in \mathcal{P}_i^a} V_j), \qquad (1)$$

where TextEnc denotes the text encoder of OmiCLIP and $| \cdot |$ denotes the concatenation operator, for joining names into a sentence.

Then, SGC adopts Delaunay Triangulation (DT) (Lee & Schachter, 1980), which can tolerate the coordinate noise and different spatial resolution, to the spot coordinates $\boldsymbol{C}$ to obtain the neighbor set $\mathcal{N}^a$ for $G^a$. Formally,

$$\mathcal{N}_i^a = \{j \mid \text{edge } (i, j) \in \text{DT}(\boldsymbol{C})\}. \qquad (2)$$

Subsequently, SGC employs the proposed GAT-based GAE (AGAE) to integrate spot embedding $\boldsymbol{b}_i^a$ with spatial information $\mathcal{N}^a$. Specifically, AGAE is composed of two GAT layers, and updates embeddings $\tilde{\boldsymbol{z}}_i^a \in \mathbb{R}^d$ as follows:

$$\tilde{\boldsymbol{z}}_i^a = \phi \underset{2d \to d}{\text{GAT}} \left( \phi \underset{d_0 \to 2d}{\text{GAT}} (\boldsymbol{b}_i^a, \mathcal{N}_i^a), \mathcal{N}_i^a \right), \qquad (3)$$

$$\underset{d_1 \to d_2}{\text{GAT}} (\hat{\boldsymbol{b}}_i, \hat{\mathcal{N}}_i) = \|_{h=1}^H \phi\left(\sum_{j \in \hat{\mathcal{N}}_i} \alpha_{i,j}^h \hat{\boldsymbol{W}}^h \hat{\boldsymbol{b}}_j\right), \qquad (4)$$

$$\alpha_{i,j}^h = \frac{\exp\left(\gamma(\boldsymbol{r}^h[\boldsymbol{s}^h \hat{\boldsymbol{b}}_i \| \boldsymbol{s}^h \hat{\boldsymbol{b}}_j])\right)}{\sum_{k \in \hat{\mathcal{N}}_i} \exp\left(\gamma(\boldsymbol{r}^h[\boldsymbol{s}^h \hat{\boldsymbol{b}}_i \| \boldsymbol{s}^h \hat{\boldsymbol{b}}_k])\right)}, \qquad (5)$$

where $\hat{\boldsymbol{W}}^h \in \mathbb{R}^{\frac{d_2}{H} \times d_1}$, $H$ denotes the predefined number of attention heads, $\alpha_{i,j}^h$ denotes learnable attention weight

from node $i$ to node $j$ in the $h$th head, and $\boldsymbol{r}^h$ and $\boldsymbol{s}^h$ denote the learnable weights for the $h$th head, respectively. Symbols $\phi$ and $\gamma$ denote the ELU and LeakyReLU activation functions, respectively. Compared to Graph Convention Network (GCN)-based GAE commonly adopted in prior studies (Li et al., 2023; Xu et al., 2024), AGAE incorporates GAT to capture the relative importance of neighboring nodes, effectively mitigating node over-smoothing. Please see Appendix A for details of the AGAE training procedure.

Finally, SGC constructs the spot graph $G^s = (\boldsymbol{B}^T, \mathcal{N}^s)$. Here, the node feature $\boldsymbol{B}^T$ is the transpose of the original matrix $\boldsymbol{B}$, and will be denoised later by the FAE associated with the spot graph (see Section 3.3 for more details). The neighbor set $\mathcal{N}_i^s$ for the $i$th node comprises the top-$k_s$ nodes whose embeddings have the highest cosine similarities to $\tilde{\boldsymbol{z}}_i^a$, defined as follows:

$$\mathcal{N}_i^s = \{j \mid A_{i,j}^s = 1\}, \qquad (6)$$

$$A_{i,j}^s = \begin{cases} 1, & \text{if } j \in \{k \mid \underset{k \in \{1, \cdots, M\}, \, k \neq i}{\arg \text{top}(k_s)} \cos(i, k)\}, \\ 0, & \text{otherwise}, \end{cases} \qquad (7)$$

$$\cos(i, k) = \frac{\tilde{\boldsymbol{z}}_i^{a\top} \tilde{\boldsymbol{z}}_k^a}{\|\tilde{\boldsymbol{z}}_i^a\| \|\tilde{\boldsymbol{z}}_k^a\|}, \qquad (8)$$

where $\| \cdot \|$ denotes the $L_2$-norm, $k_s$ denotes the predefined number of neighbors, and $i, j \in \{1, 2 \cdots, M\}$. By modeling the similarity relationship of spots that integrate multimodal semantics and spatial information, the spot graph captures biologically meaningful relationships beyond mere physical proximity.

### 3.2. Gene Graph Construction (GGC) Module

To construct the gene graph $G^g$, DeepGFT (Sun et al., 2025b) employed Pearson correlation to quantify gene co-expression, thereby analyzing the relationships among genes from the expression matrix $\boldsymbol{B}$. However, gene relationships extend beyond simple co-expression, and real biological

functional relationships often involve complex, nonlinear relationships, e.g., path sharing and functional dependency (Li et al., 2025a; Yan et al., 2024).

To better capture nonlinear relationships when constructing the neighbor set $\mathcal{N}^g$ for the gene graph $G^g = (\boldsymbol{B}, \mathcal{N}^g)$, GGC incorporates GenePT (Chen & Zou, 2025), a large genomic LM pre-trained on the National Center for Biotechnology Information (NCBI) database, to model gene relationships. Notably, as the most comprehensive and widely used repository of gene descriptions, NCBI contains descriptions for nearly all genes, spanning gene functions, pathways, and structural domains. Although a small number of gene descriptions may be missing from NCBI, this does not affect GenePT's performance. Hence, GenePT embeds genes into a semantic space that places functionally related genes (e.g., those involved in the same signaling cascade or sharing regulatory dependencies) closer together, capturing biologically meaningful nonlinear relationships beyond co-expression. Moreover, GenePT provides a Logistic Regression (LR) predictor that enables direct inference of the interactions among genes without additional training. Specifically, the embedding of the $i$th gene $\boldsymbol{t}_i$ ($i \in \{1, \cdots, N\}$) is defined as follows:

$$\boldsymbol{t}_i = \text{GenePT}(\text{text}_i), \qquad (9)$$

where $\text{text}_i$ is text description of gene $i$ on NCBI. Subsequently, GGC adopts the provided LR model to predict the gene interaction probability $p_{i,j}$, as follows:

$$p_{i,j} = \frac{1}{1 + \exp\left(-\left(\boldsymbol{s}(\boldsymbol{t}_i \| \boldsymbol{t}_j) + \boldsymbol{r}\right)\right)} \in [0,1], \qquad (10)$$

where $|\cdot|$ denotes the concatenation operator, and $\boldsymbol{s}$ and $\boldsymbol{r}$ denote the weight vector and bias term of LR, respectively.

In addition, GGC incorporates the co-expression relationship to construct the gene graph, given the input-specific nature of SRT data (i.e., the relationship between two genes may vary across tissue microenvironments). Specifically, let $\mathcal{P}_i^g$ denote the set of top-$k_g$ genes most correlated with the $i$th gene in terms of co-expression, defined as follows:

$$\mathcal{P}_i^g = \{j \mid \underset{k \in \{1, \cdots, N\}, k \neq i}{\arg \text{top}(k_g)} \text{ Pearson}(\boldsymbol{b}_i^g, \boldsymbol{b}_k^g)\}, \qquad (11)$$

where $\boldsymbol{b}_i^g = \boldsymbol{B} \cdot e_i$ and $e_i \in \mathbb{R}^{N \times 1}$ denotes the $i$th standard basis vector. Then, to construct the neighbor set $\mathcal{N}^g$, we define genes $i$ and $j$ as adjacent if and only if $j \in \mathcal{P}_i^g$ and their interaction probability $p_{i,j}$ exceeds $\tau$. Formally,

$$\mathcal{N}_i^g = \{j \mid A_{i,j}^g = 1\}, \qquad (12)$$

$$A_{i,j}^g = \begin{cases} 1, & \text{if } j \in \mathcal{P}_i^g \land p_{i,j} > \tau, \\ 0, & \text{otherwise,} \end{cases} \qquad (13)$$

where $i, j \in \{1, \cdots, N\}$. This procedure ensures that $\mathcal{N}^g$ captures both the nonlinear relationships learned by GenePT and the co-expression relationships in SRT data. Finally, analogous to the spot graph, GGC adopts the original matrix $\boldsymbol{B}$ as the node feature of the gene graph $G^g = (\boldsymbol{B}, \mathcal{N}^g)$, which will be denoised by the FAE associated with the gene graph (see Section 3.3 for more details).

### 3.3. The Architecture of EGAE

To perform element-wise denoising of SRT data, EGAE employs two GAT-based Feature Autoencoders (FAEs) to separately denoise the spot graph $G^s = (\boldsymbol{B}^T, \mathcal{N}^s)$ and gene graph $G^g = (\boldsymbol{B}, \mathcal{N}^g)$. Each FAE encoder comprises two sequential blocks, and each block consists of a GAT layer followed by a linear layer. The node embeddings $\tilde{\boldsymbol{z}}_i^s$, $\tilde{\boldsymbol{z}}_j^g \in \mathbb{R}^{2d}$ ($i \in \{1, \cdots, M\}, j \in \{1, \cdots, N\}$) for the spot and gene graphs are computed as follows:

$$\tilde{\boldsymbol{z}}_i^s = \boldsymbol{W}_2^s\left(\phi \underset{8d \to 16d}{\text{GAT}}\left(\phi \boldsymbol{W}_1^s(\phi \underset{N \to 32d}{\text{GAT}}(\hat{\boldsymbol{b}}_i^s, \mathcal{N}_i^s))\right), \mathcal{N}_i^s\right), \quad (14)$$

$$\tilde{\boldsymbol{z}}_j^g = \boldsymbol{W}_2^g\left(\phi \underset{8d \to 16d}{\text{GAT}}\left(\phi \boldsymbol{W}_1^g(\phi \underset{M \to 32d}{\text{GAT}}(\hat{\boldsymbol{b}}_j^g, \mathcal{N}_j^g))\right), \mathcal{N}_j^g\right), \quad (15)$$

where $\hat{\boldsymbol{b}}_i^s = \boldsymbol{B}^T \cdot e_i, \hat{\boldsymbol{b}}_j^g = \boldsymbol{B} \cdot e_j$. The weight matrices $\boldsymbol{W}_2^s$, $\boldsymbol{W}_2^g \in \mathbb{R}^{2d \times 16d}$, and $\boldsymbol{W}_1^s, \boldsymbol{W}_1^g \in \mathbb{R}^{8d \times 32d}$.

Subsequently, two FAE decoders, mirroring the encoders' two-block architecture, yield the spot-wise denoised expression matrix $\boldsymbol{B}^s$ and the gene-wise denoised expression matrix $\boldsymbol{B}^g$. Specifically, for the $i$th node, $\boldsymbol{b}_i^s \in \mathbb{R}^N$ and $\boldsymbol{b}_j^g \in \mathbb{R}^M$ are computed as follows:

$$\boldsymbol{b}_i^s = \rho \underset{8d \to N}{\text{GAT}}\left(\phi \boldsymbol{W}_4^s(\phi \underset{16d \to 32d}{\text{GAT}}(\phi \boldsymbol{W}_3^s \tilde{\boldsymbol{z}}_i^s, \mathcal{N}_i^s)), \mathcal{N}_i^s\right), \quad (16)$$

$$\boldsymbol{b}_j^g = \rho \underset{8d \to M}{\text{GAT}}\left(\phi \boldsymbol{W}_4^g(\phi \underset{16d \to 32d}{\text{GAT}}(\phi \boldsymbol{W}_3^g \tilde{\boldsymbol{z}}_j^g, \mathcal{N}_j^g)), \mathcal{N}_j^g\right), \quad (17)$$

where $\boldsymbol{W}_3^s, \boldsymbol{W}_3^g \in \mathbb{R}^{16d \times 2d}$, $\boldsymbol{W}_4^s, \boldsymbol{W}_4^g \in \mathbb{R}^{8d \times 32d}$, and $\rho$ denotes the RELU function. Finally, EGAE fuses $\boldsymbol{B}^s$ and $\boldsymbol{B}^g$ with the Element-wise Weighting Addition (EWA) to produce the final denoised expression matrix $\hat{\boldsymbol{B}}$ as follows:

$$\overline{B}_{i,j} = \boldsymbol{W}_{i,j}(B_{i,j}^s + B_{j,i}^g), \qquad (18)$$

$$\hat{\boldsymbol{B}} = \sigma \overline{\boldsymbol{B}} \times \boldsymbol{B}^s + (1 - \sigma \overline{\boldsymbol{B}}) \times \boldsymbol{B}^{gT}, \qquad (19)$$

where $\boldsymbol{W} \in \mathbb{R}^{M \times N}$ and $\sigma$ denotes the sigmoid function. See Appendix A for details of the EGAE training procedure.

## 4. Experimental Results

This section first benchmarks SpaEF against eight SRT data denoising methods, including STAGATE (Dong & Zhang, 2022), Sprod (Wang et al., 2022), Smoother (Su et al., 2023),

*Table 1.* RMSE values of various denoising methods across different mask proportions

| Method | Prop.=5%↓ | Prop.=10% ↓ | Prop.=15%↓ | Prop.=20% ↓ | Prop.=30%↓ | Prop.=50% ↓ |
|---|---|---|---|---|---|---|
| Sprod (Nat. Methods'22) | 0.4103±0.0028 | 0.4155±0.0016 | 0.4232±0.0008 | 0.4327±0.0018 | 0.4509±0.0002 | 0.4949±0.0007 |
| STAGATE (Nat. Commun.'22) | 0.3739±0.0018 | 0.3759±0.0015 | 0.3817±0.0022 | 0.3882±0.0019 | 0.4074±0.0024 | 0.4643±0.0008 |
| Smoother (Genome Biol.'23) | 0.6600±0.0009 | 0.6600±0.0008 | 0.6599±0.0003 | 0.6597±0.0004 | 0.6596±0.0004 | 0.6598±0.0001 |
| SEDR (Genome Med.'24) | 1.0653±0.0021 | 1.0640±0.0019 | 1.0642±0.0026 | 1.0657±0.0034 | 1.0633±0.0048 | 0.9972±0.0125 |
| DUSTED (AAAI'25) | 0.3833±0.0012 | 0.3849±0.0028 | 0.3878±0.0032 | 0.3953±0.0043 | 0.4133±0.0051 | 0.4664±0.0051 |
| DeepGFT (Genome Biol.'25) | 0.3931±0.0014 | 0.3976±0.0011 | 0.4060±0.0012 | 0.4135±0.0017 | 0.4342±0.0020 | 0.4815±0.0044 |
| **SpaEF (ours)** | 0.3664±0.0004 | 0.3675±0.0005 | 0.3701±0.0011 | 0.3760±0.0042 | 0.3837±0.0007 | 0.4172±0.0034 |

*Table 2.* PCC values of various denoising methods across different mask proportions

| Method | Prop.=5%↑ | Prop.=10%↑ | Prop.=15%↑ | Prop.=20%↑ | Prop.=30%↑ | Prop.=50%↑ |
|---|---|---|---|---|---|---|
| Sprod (Nat. Methods'22) | 0.7016±0.0047 | 0.7021±0.0018 | 0.7015±0.0013 | 0.6983±0.0032 | 0.6991±0.0017 | 0.6953±0.0011 |
| STAGATE (Nat. Commun.'22) | 0.7602±0.0007 | 0.7595±0.0018 | 0.7595±0.0006 | 0.7591±0.0006 | 0.7575±0.0003 | 0.7528±0.0002 |
| SEDR (Genome Med.'24) | 0.2600±0.0014 | 0.2588±0.0006 | 0.2569±0.0011 | 0.2524±0.0016 | 0.2457±0.0018 | 0.2453±0.0016 |
| DUSTED (AAAI'25) | 0.7453±0.0008 | 0.7463±0.0006 | 0.7469±0.0006 | 0.7474±0.0008 | 0.7485±0.0006 | 0.7496±0.0004 |
| DeepGFT (Genome Biol.'25) | 0.7305±0.0007 | 0.7270±0.0006 | 0.7228±0.0001 | 0.7180±0.0003 | 0.7068±0.0008 | 0.6746±0.0004 |
| **SpaEF (ours)** | 0.7679±0.0011 | 0.7679±0.0016 | 0.7669±0.0016 | 0.7607±0.0041 | 0.7581±0.0030 | 0.7333±0.0059 |

DUSTED (Zhu et al., 2025a), SEDR (Xu et al., 2024), DeepGFT (Sun et al., 2025b), Spotscape (Oh et al., 2025), and scGPT-spatial (Wang et al., 2025) on four real-world datasets from different platforms and tissues to assess the performance of SpaEF, then conducts ablation studies to validate the effectiveness of all proposed modules. Appendices B~G provide the experimental setups (details about baselines and datasets), spatial domain identification and clustering results, correlation analysis between PDGFRA and PRF1 genes, computational complexity analysis, robustness and structural validity of the constructed spot and gene graphs, and sensitivity analyses of hyperparameters, respectively. The code of SpaEF is available online[1].

### 4.1. Comparison on the HOCWT Dataset

In this subsection, we mask portions of expression data and then recover them using different denoising methods to evaluate performance. Following (Wang et al., 2022), we adopt the expression matrix from the widely used, real-world HOCWT dataset. Notably, the simulated datasets (e.g., sc-Cube (Qian et al., 2024), scDesign3 (Song et al., 2024)) entail underlying assumptions about the expression distribution; consequently, treating such simulations as ground truth can bias or otherwise distort evaluation. Therefore, we perform masking experiments using only the real-world HOCWT dataset. For each masking proportion (5%, 10%, 15%, 20%, 30%, and 50%), we randomly mask the expression matrix, using ten distinct random seeds to assess

stability. We then apply each denoising method and evaluate its accuracy by comparing the recovered values to the true values at the masked locations. For fair comparison, we compute both the Root Mean Squared Error (RMSE) and the Pearson Correlation Coefficient (PCC). RMSE measures the absolute deviation between the recovered and ground-truth values, whereas PCC assesses whether the recovered values preserve the overall expression patterns (i.e., relative ordering or co-variation) of the true masked values.

As presented in Tables 1 and 2, SpaEF consistently achieves the lowest RMSE across all mask proportions and attains the highest PCC value in most settings. Even under high masking proportions (e.g., 30% and 50%), SpaEF retains competitive PCC relative to the baselines such as STAGATE and DeepGFT, highlighting its robustness against extreme data sparsity. In addition, we compare SpaEF to DeepGFT using the Wilcoxon signed-rank test, which indicates statistically significant differences in both RMSE and PCC. These results demonstrate that SpaEF not only achieves superior accuracy in recovering masked expression values but also preserves relative expression patterns that are crucial for downstream analyses. The high-level performance of SpaEF can be attributed to its ability to effectively leverage spatial and expression information, thereby balancing recovering accuracy (low RMSE) and structural preservation (high PCC). In addition, this robustness under severe masking makes SpaEF a promising method for denoising SRT datasets, which often suffer from high dropout rates and missing signals.

---

[1]https://github.com/Zekuan-Shang/SpaEF

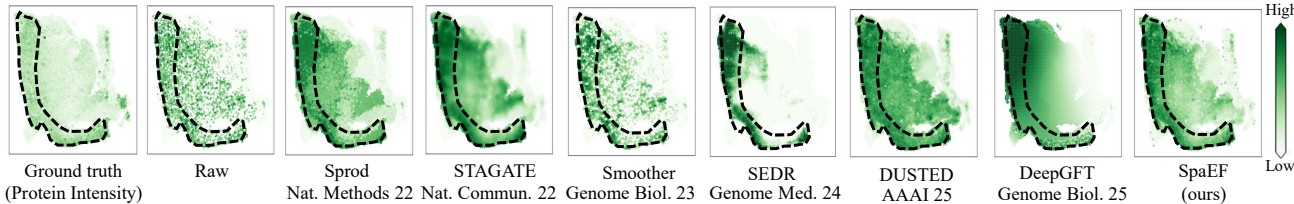

Ground truth (Protein Intensity)    Raw    Sprod Nat. Methods 22    STAGATE Nat. Commun. 22    Smoother Genome Biol. 23    SEDR Genome Med. 24    DUSTED AAAI 25    DeepGFT Genome Biol. 25    SpaEF (ours)

*Figure 3.* Visualization of the spatial distribution of PCNA gene expression denoised by different methods on the HGBM dataset. The ground truth refers to the spatial distribution of the PCNA protein. Raw refers to the spatial distribution of PCNA gene expression before denoising. The region enclosed by the black dashed line denotes the expected gene expression region.

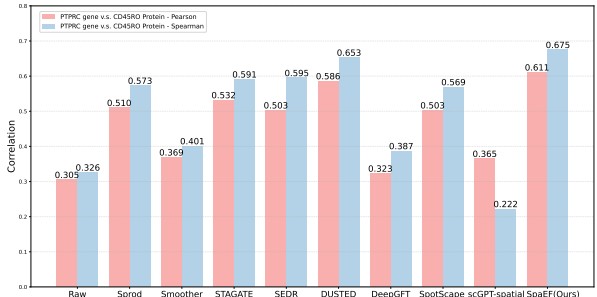

*Figure 4.* Performance comparison of various denoising methods on the HGBM dataset, quantified by PCC and SCC between PCNA protein intensity and PCNA expression.

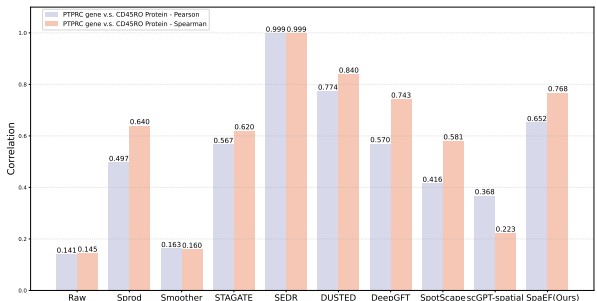

*Figure 5.* Performance comparison of various denoising methods on the HGBM dataset, quantified by PCC and SCC between CD44 and MYC gene expressions.

### 4.2. Comparison on the HGBM Dataset

Following the prior studies (Wang et al., 2022; Zhu et al., 2025a), we assess the performance of different denoising methods on recovering the positive correlation between gene expression and corresponding protein intensity on the HGBM dataset. This experiment enables a direct assessment of each method's effectiveness at recovering real biological signals and reducing noise-induced distortion. Specifically, we define protein intensity as a constant derived from the mean pixel value within each spot region on the immunofluorescence (IF) image, and gene expression as the corresponding values from the raw or denoised expression matrices. Among various gene–protein pairs, we focus on the correlation of the PCNA gene and PCNA protein, because the prior study (Ong et al., 2017) has shown that PCNA gene expres-

sion is significantly elevated in GBM tissues compared to low-grade gliomas and normal brain specimens, making it a particularly informative benchmark. Figure 4 presents the Pearson Correlation Coefficient (PCC) and Spearman Correlation Coefficient (SCC) values between PCNA protein intensity and denoised PCNA expression across different methods. The results show that SpaEF yields markedly higher PCC and SCC values than all other methods, underscoring its ability to faithfully preserve biologically meaningful signals embedded in noisy SRT data.

Moreover, Figure 3 illustrates the PCNA protein intensity (ground truth) with both raw and denoised PCNA gene expressions. The ground truth refers to the spatial distribution of the PCNA protein, extracted from the corresponding IF image. In the raw expression data, the distinction between high- and low-expression regions is unclear. DeepGFT oversmooths the spatial distribution of PCNA, failing to preserve the expression contrast. In contrast, our SpaEF produces the PCNA expression that closely matches the ground truth, effectively capturing the high-expression band along the tissue's left side and the low-expression region at its center. Although DUSTED achieves solid correlation scores (see Figure 4), it does not recover the central low-expression region as effectively as SpaEF. The superior framework of SpaEF enables it to reconstruct PCNA expression that closely tracks the PCNA protein intensities, thereby more accurately reflecting true cellular functional states.

Finally, following the prior study (Wang et al., 2022), we assess whether the expression correlations between genes in the denoised outputs align with established biological knowledge. Specifically, we select genes CD44 and MYC as benchmark genes, because these two genes have a significant positive correlation (Wang et al., 2018), whereas this correlation is not significant in the raw data. As shown in Figure 5, SpaEF attains the third-highest PCC and SCC values compared to other methods. Nevertheless, both PCC and SCC values exceed 0.5, indicating that SpaEF successfully captures the expected positive association between CD44 and MYC from noisy SRT data. Although SEDR achieves the highest PCC and SCC for this gene pair, it appears to systematically overestimate the correlation across gene pairs. For example, it reports a strong, likely spurious correlation

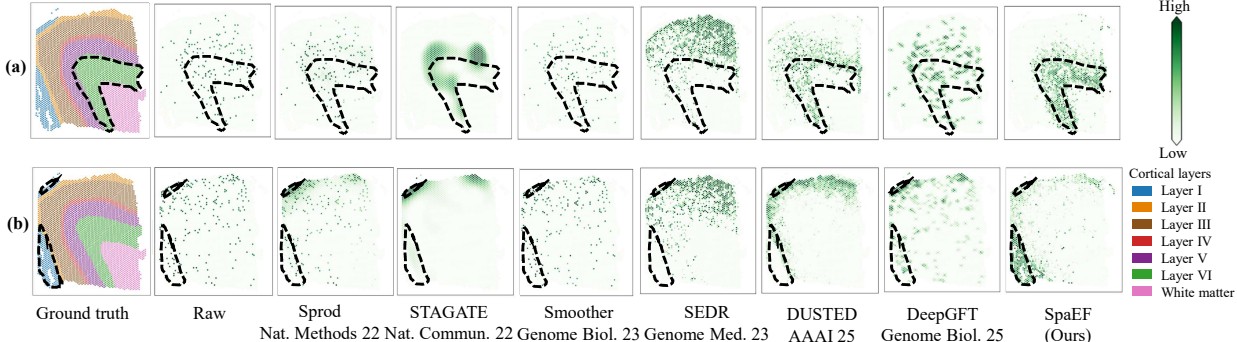

*Figure 6.* Visualization of the spatial distribution of FOXP2 (a) and RELN (b) gene expressions denoised by different methods on the DLPFC dataset. The ground truth refers to the manual annotation layers and white matter. Raw refers to the spatial distribution of gene expression before denoising. The region enclosed by the black dashed line denotes the expected gene expression region.

between PDGFRA and PRF1, which are not expected to exhibit high co-expression (see Appendix D), indicating limited reliability. By contrast, SpaEF maintains appropriately low correlation for PDGFRA and PRF1, demonstrating its ability to distinguish biological signal from noise and to reserve true biological relationships.

### 4.3. Comparison on the DLPFC Dataset

Following the prior study (Zhu et al., 2025a), we evaluate SpaEF's ability to recover gene spatial expression patterns using the DLPFC dataset. Specifically, we select FOXP2 and RELN, which are known to be predominantly expressed in layer VI and layer I of the DLPFC, respectively, with minimal expression elsewhere (Qian et al., 2025; Camacho et al., 2014). Ground truth annotations are drawn from the study (Maynard et al., 2021). Figure 6 shows the gene spatial expressions after denoising by various methods. As shown in Figure 6(a), our SpaEF more accurately recovers the concentrated expression of FOXP2 in layer VI, whereas the results from other methods, including the raw data, appear more dispersed. In addition, other methods fail to effectively recover the RELN expression in the lower-left region of layer I, as shown in Figure 6(b), while our SpaEF successfully restores it. This suggests that SpaEF preserves fine-grained spatial variation in gene expression, thereby maintaining heterogeneous expression patterns across different cellular microenvironments. We then evaluate the performance of SpaEF and baselines on the other two downstream tasks: spatial domain identification and clustering, in Appendix C. The results indicate that SpaEF can better preserve the domain-level or cluster-level structures in SRT data.

### 4.4. Comparison on the HDHBC Dataset

To stress-test these methods on extremely sparse data, we mirrored the HGBM evaluation by measuring the correla-

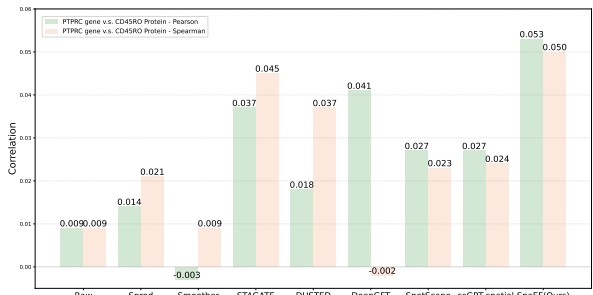

*Figure 7.* Performance comparison of various denoising methods on the HDHBC e, quantified by PCC and SCC between CD45RO protein intensity and PTPRC expression.

tion between denoised expression and protein intensity. We chose HDHBC for its 98.66% zero-expression rate, presenting a much greater challenge than other SRT datasets, which usually have sparsity rates under 90%. The selected correlation is that between CD45RO protein intensity and PTPRC gene expression. Because CD45RO is a transmembrane glycoprotein encoded by the PTPRC gene and is widely used as an indicator of tumor-infiltrating immune cells (Ahmadvand et al., 2019). As shown in Figure 7, none of the methods achieve a gene–protein correlation coefficient above 0.1, owing to the extreme sparsity of the HDHBC expression matrix. Nevertheless, because protein intensity values are constant, which is fixed and shared across all methods, the observed differences in gene-protein correlation directly reflect the effectiveness of the denoising. Under this setting, SpaEF demonstrates superior performance over competing methods. In particular, compared with STAGATE, the second-best method, SpaEF achieves a 40.54% improvement in PCC and an 11.11% improvement in SCC.

### 4.5. Ablation Studies

We conduct ablation studies on the HGBM dataset to assess the effectiveness of key designs and modules of SpaEF, and the experimental results are presented in Table 3. Specifi-

*Table 3.* Ablation study on different SpaEF configurations

| Method | PCC (Gap) | SCC (Gap) |
|---|---|---|
| w/o OmiCLIP (PCA) | 0.531 (-13.09%) | 0.605 (-10.37%) |
| w/o OmiCLIP (scGPT) | 0.569 (-6.87%) | 0.639 (-5.33%) |
| w/o OmiCLIP (Geneformer) | 0.496 (-18.82%) | 0.557 (-17.48%) |
| w/o GenePT (co-only) | 0.540 (-11.62%) | 0.614 (-9.04%) |
| w/o GenePT (scGPT) | 0.548 (-10.31%) | 0.620 (-8.15%) |
| w/o GenePT (Geneformer) | 0.543 (-11.12%) | 0.616 (-8.74%) |
| w/o AGAE | 0.559 (-8.51%) | 0.633 (-6.22%) |
| w/o Gene graph | 0.539 (-11.78%) | 0.615 (-8.89%) |
| w/o EWA (matrix-wise) | 0.573 (-6.22%) | 0.646 (-4.30%) |
| w/o EWA (cross-attention) | 0.555 (-9.16%) | 0.622 (-7.85%) |
| SpaEF | **0.611 (-)** | **0.675 (-)** |

cally, w/o OmiCLIP (PCA/scGPT/Geneformer) refer to the settings that replaces OmiCLIP's text encoder with PCA or with scGPT (Cui et al., 2024) / Geneformer (Theodoris et al., 2023) (scGPT and Geneformer are two single-modality LMs focus on representation learning) to encode spot embeddings; w/o GenePT (co-only) refers to the setting that only adopts the co-expression relationship for constructing the gene graph, w/o GenePT (scGPT/Geneformer) refers to the setting that substitutes GenePT with scGPT or Geneformer, and the gene interaction probabilities are computed using cosine similarity rather than the LR predictor provided by GenePT; w/o AGAE refers to the setting that replaces the GAT-based AGAE with a GCN-based GAE adopted in scGNN (Wang et al., 2021), w/o Gene Graph refers to the setting that only the spot graph is used for denoising, and w/o EWA (matrix-wise) refers to the setting that replaces EWA with matrix-wise weighting addition adopted in DeepGFT(Sun et al., 2025b) and w/o EWA (cross-attention) refers to the setting that fuse the spot graph and gene graph taking spot-wise feature as query, gene-wise feature as key/value. The Gap is defined as 1-(performance of the ablated model)/(performance of the intact SpaEF), measuring the relative performance drop from the intact SpaEF. The experimental results show that ablating any individual component significantly decreases PCC and SCC between PCNA gene expression and protein intensity on the HGBM dataset, demonstrating that each component contributes to the performance of SpaEF.

## 5. Conclusion

We propose SpaEF, an element-wise SRT denoising method comprising three modules, namely SGC, GGC, and EGAE. SGC leverages OmiCLIP to extract spot semantic features, and then employs the proposed AGAE to integrate these features with spatial coordinates for spot graph construction. To construct the gene graph, GGC exploits GenePT to capture nonlinear gene relationships that complement the co-expression relationship. Finally, EGAE fuses the spot

and gene graphs to achieve element-wise denoising. Experiments validate that SpaEF outperforms SOTA denoising methods and exhibits strong robustness across tasks.

## Limitations

We acknowledge that, although SpaEF achieves the best denoising results on the HDHBC dataset compared to other methods, the absolute performance metrics remain poor, indicating that SpaEF still cannot effectively denoise extremely sparse datasets.

## Acknowledgements

This work was supported by the Jilin Provincial Department of Science and Technology Project under Grant 20230201083GX.

## Impact Statement

This paper presents work whose goal is to advance the field of Machine Learning. There are many potential societal consequences of our work, none which we feel must be specifically highlighted here.

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

## A. Training procedure of SpaEF

Since we directly adopt the pre-trained OmiCLIP and GenePT to construct spot and gene graphs, SpaEF contains only two trainable networks, namely AGAE and EGAE. Following the prior study scGNN (Wang et al., 2021), we train these two networks separately, first AGAE, then EGAE. To train AGAE, we adopt the cross-entropy loss $\mathcal{L}_{cs}$, which minimizes the difference between the original adjacency matrix $\boldsymbol{A}^a$ of the spatial graph and the reconstructed adjacency matrix $\hat{\boldsymbol{A}}^a$, computed as the inner product of the spatial graph node embeddings $\tilde{\boldsymbol{z}}_i^a$. Formally,

$$\mathcal{L}_{cs}(\boldsymbol{A}^a, \hat{\boldsymbol{A}}^a) = -\frac{1}{M \times M} \sum_{i=1}^M \sum_{j=1}^M \left( A_{i,j}^a * \log(\hat{A}_{i,j}^a) + (1 - A_{i,j}^a) * \log(1 - \hat{A}_{i,j}^a) \right), \tag{20}$$

$$A_{i,j}^a = \begin{cases} 1, & \text{if } j \in \mathcal{N}_i^a, \\ 0, & \text{otherwise,} \end{cases} \tag{21}$$

$$\hat{A}_{i,j}^a = \sigma(\tilde{\boldsymbol{z}}_i^a \times \tilde{\boldsymbol{z}}_j^a). \tag{22}$$

In addition, we employ the Adam optimizer for training. During optimization, gradients are backpropagated to update the learnable parameters by minimizing $\mathcal{L}_{cs}$, thereby enabling AGAE to learn informative spot embeddings.

To train EGAE, following prior art DUSTED (Zhu et al., 2025a) in this field, we employ the Zero-Inflated Negative Binomial (ZINB) (Zhao et al., 2022) loss function and use ADAM optimizer. This choice is motivated by the significant zero-inflation (i.e., $b_{i,j} = 0$) and overall sparsity of the expression matrix, which the ZINB distribution flexibly accommodates. Specifically, we define the loss function $\mathcal{L}_Z$ as the negative log-likelihood of the ZINB distribution over the observed matrix $\boldsymbol{B}$. Formally,

$$\mathcal{L}_Z = -\sum_{i,j} \log \left( P(b_{i,j} | \mu_{i,j}, \theta_j, \pi_{i,j}) \right) + \frac{1 \times e^{-3}}{M \cdot N} \sum_{i,j} (\mu_{i,j})^2, \tag{23}$$

where $i \in \{1, \cdots, M\}$, $j \in \{1, \cdots, N\}$, $\pi$ denotes the dropout probability, $\mu$ denotes the expected expression matrix (i.e., the denoised expression matrix $\hat{\boldsymbol{B}}$), and $\theta_j$ is the gene-specific dispersion parameter for the $j$th gene in $\boldsymbol{B}$. The probabilities of $b_{i,j} = 0$ and $b_{i,j} = k$ ($k$ is a positive integer) are computed as follows:

$$\begin{aligned} P(b_{i,j} = 0) &= \pi_{ij} + (1 - \pi_{ij}) \cdot \text{NB}(0; \mu_{ij}, \theta_j), \\ P(b_{i,j} = k) &= (1 - \pi_{ij}) \cdot \text{NB}(k; \mu_{ij}, \theta_j). \end{aligned} \tag{24}$$

where $\text{NB}(\cdot)$ denotes the probability mass function of the negative binomial distribution.

## B. Experimental Setups

**Datasets and Preprocessing Methods:** To assess the performance of SpaEF, we conduct extensive experiments on a masked real-world dataset, namely HOCWT[2], and three real-world datasets, namely HGBM[3], HDHBC[4], and DLPFC (Maynard et al., 2021). The HOCWT dataset, which provides whole transcriptome analysis on human ovarian cancer tissue, comprises an expression matrix, spot coordinates, and matching immunofluorescence (IF) protein-staining images. The HGBM dataset, derived from human glioblastoma (GBM) tissue, comprises an expression matrix, spot coordinates, and matching IF protein-staining images. The HDHBC dataset, derived from human breast cancer tissue, features an extremely sparse expression matrix (zero-expression rate: 98.66%), spot coordinates, and corresponding IF protein-staining images. Additionally, section 151673 of the DLPFC dataset provides an expression matrix, spot coordinates, and expert-annotated spot types. For all datasets, we adopt a standardized preprocessing pipeline following prior studies (Wang et al., 2021; 2022): (1) discard genes expressed in fewer than 1% of spots (0.5% for HDHBC); (2) discard spots expressing fewer than 1% of genes; (3) normalize each spot's counts to counts per million (CPM) to mitigate batch effects; and (4) apply a log1P transformation to reduce skewness while preserving zeros. For the HOCWT dataset, after preprocessing, we additionally apply random masking to the expression matrix at proportions of 5%, 10%, 15%, 20%, 30%, and 50%. To ensure experimental consistency with (Liu et al., 2025), each masking level is repeated with ten independent random seeds.

---

[2]Human Ovarian Cancer: Whole Transcriptome Analysis. Stains: DAPI, Anti-PanCK, Anti-CD45, www.10xgenomics.com/datasets

[3]Visium CytAssist Gene and Protein Expression Library of Human Glioblastoma, IF (FFPE), www.10xgenomics.com/datasets

[4]Visium HD Spatial Gene Expression Library, Human Breast Cancer, IF (FFPE), www.10xgenomics.com/datasets

**Benchmarking methods:** To evaluate the performance of SpaEF, we compare it with eight SRT denoising methods, namely STAGATE (Dong & Zhang, 2022), Sprod (Wang et al., 2022), Smoother (Su et al., 2023), DUSTED (Zhu et al., 2025a), SEDR (Xu et al., 2024), DeepGFT (Sun et al., 2025b), Spotscape(Oh et al., 2025), and scGPT-spatial(Wang et al., 2025). Among them, Sprod and Smoother are graph-regularized methods, STAGATE, SEDR and Spotscape are GNN-based method that utilizes only spot-wise features, and DUSTED and DeepGFT are GNN-based methods that incorporate both spot- and gene-wise features. scGPT-spatial is an LM-based method, and we run it in zero-shot with the provided model weights.

**Evaluation Metrics:** Following prior studies (Pham et al., 2023; Wang et al., 2022; Zhu et al., 2025a), we evaluate different denoising methods according to four complementary metrics: (1) the accuracy in recovering randomly masked gene expression values measured by RMSE as Eq. (25) and PCC as Eq. (26); (2) the correlation between the denoised gene expression and protein intensity measured by PCC as Eq. (26) and SCC as Eq. (27); (3) the correlation between pairs of denoised gene expression, measured by PCC as Eq. (26) and SCC as Eq. (27); (4) the recovery of gene spatial distributions, visualized in spatial spot graph as Figure 3 and Figure 6.

*Root Mean Squared Error (RMSE):* Let $B \in \mathbb{R}^{M \times N}$ denote the original gene expression matrix and $\hat{B} \in \mathbb{R}^{M \times N}$ the denoised output. We quantify the reconstruction accuracy using the RMSE:

$$\text{RMSE}(B, \hat{B}) = \sqrt{\frac{1}{MN} \sum_{i=1}^{M} \sum_{j=1}^{N} \left( B_{i,j} - \hat{B}_{i,j} \right)^2}. \tag{25}$$

*Pearson Correlation Coefficient (PCC) and Spearman rank Correlation Coefficient (SCC):* We report both PCC and SCC to capture linear consistency as well as rank-based monotonic relationships, which is particularly important for SRT data with complex noise mechanisms. For a given gene $j$, let $b_j^g = B \cdot e_j \in \mathbb{R}^M$ denote the original or denoised gene expression vector across spots, and let $\boldsymbol{p}_j \in \mathbb{R}^M$ denote the matched protein intensity vector. In addition, PCC on the HOCWT dataset is computed by flattening the matrix into a single vector. The PCC is defined as

$$\text{PCC}(b_j^g, \boldsymbol{p}_j) = \frac{\sum_{i=1}^{M} (b_{j,i}^g - \breve{b}_j^g)(p_{j,i} - \breve{p}_j)}{\sqrt{\sum_{i=1}^{M} (b_{j,i}^g - \breve{b}_j^g)^2} \sqrt{\sum_{i=1}^{M} (p_{j,i} - \breve{p}_j)^2}}, \tag{26}$$

where $\breve{b}_j^g = \frac{1}{M} \sum_{i=1}^{M} b_{j,i}^g$ and $\breve{p}_j = \frac{1}{M} \sum_{i=1}^{M} p_{j,i}$ denote the average element or protein strength values across all spots, respectively. In addition, SCC is defined as:

$$\text{SCC}\left(b_j^{(g)}, \boldsymbol{p}_j\right) = \text{PCC}\left(\text{rank}\left(b_j^{(g)}\right), \text{rank}(\boldsymbol{p}_j)\right). \tag{27}$$

To assess whether pairwise gene relationships are preserved after denoising, we compute the PCC and SCC between pairs of denoised gene expression vectors. Given two genes $j$ and $k$, their PCC and SCC is measured as $\text{PCC}(b_j^g, b_k^g)$ and $\text{SCC}(b_j^g, b_k^g)$.

*Recovery of gene spatial distributions:* Finally, we evaluate whether spatial expression patterns are preserved after denoising by examining the spatial distribution of denoised gene expression across tissue coordinates, following the protocols in prior work Sprod (Wang et al., 2022).

**Detailed Hyperparameters:** In Table 4, we present the hyperparameters of SpaEF. We use the same settings across all datasets.

**Hardware:** We comprehensively evaluate the performance of all methods, using a computer equipped with a 22 vCPU Intel(R) Xeon(R) Platinum 8470Q CPU and a RTX PRO 6000 GPU.

## C. Spatial Domain Identification and Clustering Experiments

To further assess SpaEF on practical downstream applications, in this section, we conduct non-spatial clustering and spatial domain recognition experiments on DLPFC. For clustering, we first extract the top 2000 highly variable genes (HVGs) across spots and then reduce dimensionality using standard PCA. Following DUSTED (Zhu et al., 2025a), we apply the mclust (Scrucca et al., 2016) clustering method on the low-dimensional embeddings. For spatial domain identification, we

*Table 4.* Hyperparameters of SpaEF

| Parameter | Value |
|---|---|
| $k_e$,$k_s$,$k_g$ | 50, 10, 10 |
| $d_0$,$d$ | 768, 32 |
| $\tau$ | 0.5 |
| learning rate of EGAE | le-4 |
| learning rate of AGAE | 1e-2 |
| epochs number of EGAE | 800 |
| epochs number of AGAE | 1000 |
| the number of GAT heads $H$ | 4 |

*Table 5.* Results of Spatial Domain Identification and Clustering

| Method | Domain.ARI | Domain.ACC | Clustering.ARI | Clustering.ACC |
|---|---|---|---|---|
| raw | $0.262 \pm 0.040$ | $0.469 \pm 0.034$ | $0.040 \pm 0.010$ | $0.308 \pm 0.005$ |
| Sprod (Wang et al., 2022) | $0.274 \pm 0.032$ | $0.457 \pm 0.003$ | $0.270 \pm 0.051$ | $0.446 \pm 0.038$ |
| STAGATE (Dong & Zhang, 2022) | $0.338 \pm 0.042$ | $0.537 \pm 0.024$ | $0.282 \pm 0.037$ | $0.467 \pm 0.033$ |
| Smother (Su et al., 2023) | $0.360 \pm 0.030$ | $0.523 \pm 0.037$ | $0.199 \pm 0.023$ | $0.417 \pm 0.018$ |
| DUSTED (Zhu et al., 2025a) | $0.325 \pm 0.045$ | $0.511 \pm 0.034$ | $0.271 \pm 0.031$ | $0.488 \pm 0.028$ |
| DeepGFT (Sun et al., 2025b) | $0.306 \pm 0.021$ | $0.480 \pm 0.021$ | $0.295 \pm 0.029$ | $0.466 \pm 0.026$ |
| SpaEF (ours) | $0.353 \pm 0.020$ | $0.544 \pm 0.019$ | $0.297 \pm 0.050$ | $0.501 \pm 0.052$ |

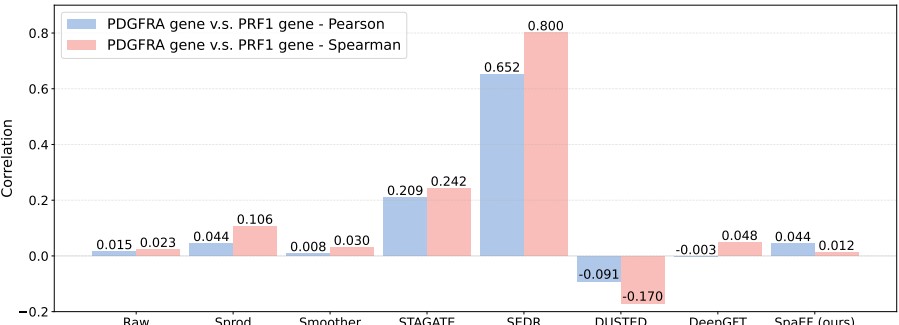

*Figure 8.* Performance comparison of various denoising methods on the HGBM dataset, quantified by PCC and SCC between PDGFRA and PRF1 gene expressions.

apply SpaGCN (Hu et al., 2021) on the expression matrix and coordinates. Every experiment is repeated five times, and results are quantified with two metrics, Adjusted Rand Index (ARI) and accuracy (ACC), as shown in Table 5. The results show that SpaEF improves domain identification and clustering quality compared to baselines, indicating better preservation of domain-level or cluster-level structures.

## D. Correlation analysis between PDGFRA and PRF1 genes

In this section, we investigate whether denoising methods can spuriously infer correlations among genes in the HGBM dataset. Specifically, we analyze the gene pair PDGFRA and PRF1, whose expressions are expected to be uncorrelated or negatively correlated, because PDGFRA is a canonical marker of GBM tumor cells (Verhaak et al., 2010), whereas PRF1 is a marker gene of immune cells (de los Rios Kobara et al., 2025). As shown in Figure 8, most methods produce low correlations between these two genes, but SEDR still yields a spuriously high correlation. This indicates that its results may lack reliability and exhibit a systematic overestimation of correlations between gene pairs on the HGBM dataset.

*Table 6.* Running time of different methods

| Method | Running Time (s) |
|---|---|
| Sprod (Wang et al., 2022) | 146.61 |
| STAGATE (Dong & Zhang, 2022) | 14.33 |
| Smoother (Su et al., 2023) | 57.85 |
| SEDR (Xu et al., 2024) | 9.65 |
| DUSTED (Zhu et al., 2025a) | 21.84 |
| DeepGFT (Sun et al., 2025b) | 324.97 |
| SpaEF (ours) | 111.40 |

*Table 7.* Robustness analysis under cell and gene graph edge perturbations on HGBM dataset

| Spot graph Drop Rate | Gene Graph Drop Rate | PCC | SCC |
|---|---|---|---|
| 0% | 0% | **0.611** | **0.675** |
| 10% | 10% | 0.539 | 0.612 |
| 10% | 20% | 0.552 | 0.632 |
| 10% | 50% | 0.543 | 0.617 |
| 20% | 10% | 0.549 | 0.627 |
| 20% | 20% | 0.531 | 0.600 |
| 20% | 50% | 0.569 | 0.636 |
| 50% | 10% | 0.553 | 0.624 |
| 50% | 20% | 0.554 | 0.628 |
| 50% | 50% | 0.547 | 0.615 |

## E. Computational Complexity

In this section, we analyze the computational complexity of SpaEF and DeepGFT. DeepGFT is a representative method that constructs both spot and gene graphs. Since OmiCLIP and GenePT are pre-trained LMs, SpaEF can efficiently extract spot-level and gene-level features with a computational complexity of $O(M + N)$, where $M = 3,493$ spots and $N = 4,647$ genes. In contrast, DeepGFT relies on a computationally intensive graph Fourier transform, with a complexity of $O(M \cdot N \cdot D)$, where $D = 1,500$ spectral signals. This efficiency advantage is also reflected in practice. As shown in Table 6, we compare SpaEF with all six baseline methods in running time on the HOCWT dataset. The experimental results show that SpaEF not only runs faster than DeepGFT (111.40 s vs. 324.97 s in total denoising time) but also achieves significantly better performance. For the other baselines, SpaEF may incur a higher denoising time, but it achieves better denoising performance (see Section 4). Since SRT denoising is not a real-time task and minute-level runtime is typically acceptable in bioinformatics pipelines, SpaEF offers a practical trade-off between efficiency and denoising quality.

## F. Robustness and Structural Validity of Constructed Spot and Gene Graphs

To better understand the robustness and structural validity of the spot and gene graphs constructed by SpaEF, we conduct a series of analyses from these complementary perspectives: robustness to graph perturbations, structural properties of spot graphs induced by different feature extractors, and biological plausibility of gene graphs validated against external knowledge bases.

To evaluate the robustness of SpaEF to perturbations in graph topology, we conduct stress tests on the HGBM dataset by randomly dropping edges from both the spot graph and the gene graph. Specifically, we independently remove 10%, 20%, and 50% of edges from each graph, resulting in nine perturbation settings. As shown in Table 7, SpaEF remains resilient to structural noise, maintaining stable performance on PCNA expression recovery across all perturbation levels (PCC ranging from 0.531 to 0.569 and SCC from 0.600 to 0.636). These results indicate that SpaEF does not rely on fragile graph connectivity patterns and is robust to substantial graph sparsification, which is important for real-world spatial transcriptomics data where graph construction is often noisy or imperfect.

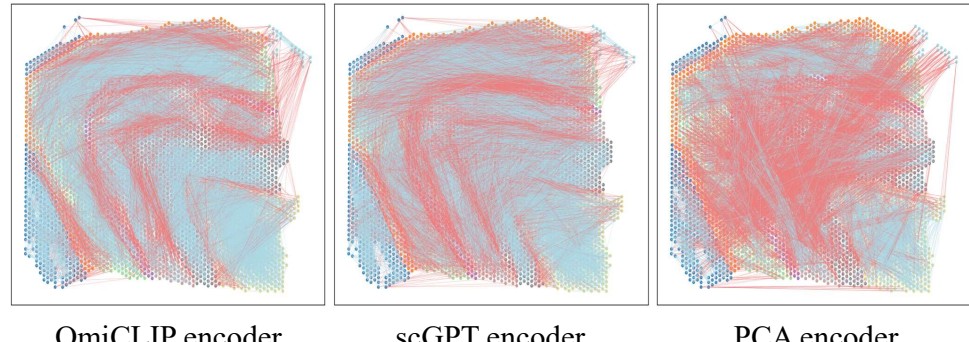

|  |  |  |
| :---: | :---: | :---: |
| OmiCLIP encoder | scGPT encoder | PCA encoder |

*Figure 9.* Visualization of spot graphs constructed using OmiCLIP, scGPT, and PCA on the DLPFC dataset.

*Table 8.* Modularity scores of different spot graphs

| Ablation Method | Modularity Score |
| :--- | :---: |
| OmiCLIP | **0.543** |
| scGPT | 0.491 |
| PCA | 0.405 |

*Table 9.* Functional precision of Gene Graph

| Gene Graph | Precision |
| :--- | :---: |
| Co-expression only | 0.94% |
| GenePT only | 0.95% |
| GenePT&Co-expression (the adopted configuration) | 1.46% |

In addition, we analyze how different spot feature extractors affect the structural properties of the resulting spot graphs to validate that OmiCLIP can mitigate spurious similarity biases in the spot graph. Specifically, we compare OmiCLIP embeddings with scGPT embeddings and PCA features by visualizing the spot graphs produced by these three different spot encoders. scGPT (Cui et al., 2024) is a foundation model pre-trained solely on scRNA-seq data, yielding expression-only spot features without external semantic or multimodal (image) information. PCA provides a linear, variance-based baseline that captures dominant expression trends but cannot model higher-order or semantic relationships. For each graph ($k_s = 10$), we color the edge in $\mathcal{N}^s$ using two labels: the blue edge indicates that its two endpoint spots belong to the same cluster according to the clustering annotations from (Maynard et al., 2021), whereas the red edge indicates that its two endpoint spots belong to different clusters. Because spots within the same cluster are expected to share transcriptional profiles, a higher proportion of blue edges indicates that transcriptional similarity relationships are better preserved and, consequently, superior performance of the spot encoder. As shown in Figure 9, OmiCLIP and scGPT can better capture cluster-consistent relationships than the conventional PCA, which fails to preserve these relationships and exhibits a higher proportion of inter-cluster (red) edges. Based on the manual cluster annotations of (Maynard et al., 2021), the accuracies (the number of blue edges divided by the total number of spot graph edges) of OmiCLIP, scGPT, and PCA are approximately 72.08%, 66.93%, and 58.49%, respectively. Notably, both OmiCLIP and scGPT exhibit instability in preserving boundaries near cluster margins, reflecting the inherent difficulty of graph construction in transitional regions. Nevertheless, OmiCLIP yields the clearest separation among the three methods. This analysis of the constructed graphs demonstrates that the edges of our spot graph are biologically meaningful, indicating that our design effectively aligns LM-derived embeddings with biological pathways.

To further quantify differences among the spot graphs constructed with OmiCLIP, scGPT, and PCA, following Stardust (Avesani et al., 2022), we compute the modularity of each graph on the DLPFC dataset. Modularity measures how well a graph can be partitioned into densely connected communities with sparse inter-community links; higher values indicate stronger alignment between the graph structure and the underlying cluster organization. As shown in Table 8, the modularity results are consistent with the visualizations in Figure 9, demonstrating OmiCLIP's superior ability to construct biologically meaningful spot graphs. Overall, OmiCLIP yields higher-modularity graphs with clearer community structure,

*Table 10.* Sensitivity analyses for different values of $k_s$ and $k_g$

| $k_s$ | $k_g$ | PCC | SCC | Clustering.ARI | Clustering.ACC |
|---|---|---|---|---|---|
| 5 | 5 | 0.554 | 0.624 | 0.182 | 0.417 |
| 5 | 10 | 0.561 | 0.626 | 0.192 | 0.417 |
| 5 | 15 | 0.551 | 0.615 | 0.187 | 0.424 |
| 10 | 5 | 0.574 | 0.651 | 0.240 | 0.453 |
| 10 | 10 | **0.611** | **0.675** | **0.297** | **0.501** |
| 10 | 15 | 0.557 | 0.632 | 0.239 | 0.447 |
| 15 | 5 | 0.543 | 0.620 | 0.251 | 0.469 |
| 15 | 10 | 0.575 | 0.648 | 0.230 | 0.443 |
| 15 | 15 | 0.538 | 0.616 | 0.261 | 0.467 |

*Table 11.* Sensitivity analyses for different $k_e$ values

| $k_e$ | PCC | SCC | Clustering.ARI | Clustering.ACC |
|---|---|---|---|---|
| 15 | 0.569 | 0.644 | 0.253 | 0.478 |
| 25 | 0.532 | 0.609 | 0.234 | 0.465 |
| 50 | **0.611** | **0.675** | **0.297** | **0.501** |
| 65 | 0.548 | 0.625 | 0.268 | 0.480 |
| 75 | 0.565 | 0.635 | 0.240 | 0.465 |

*Table 12.* Sensitivity analyses for different threshold $\tau$ values

| $\tau$ | PCC | SCC | Clustering.ARI | Clustering.ACC |
|---|---|---|---|---|
| 0.4 | 0.545 | 0.614 | 0.251 | 0.462 |
| 0.5 | **0.611** | **0.675** | **0.297** | **0.501** |
| 0.6 | 0.540 | 0.611 | 0.274 | 0.485 |

suggesting that pre-trained multimodal LMs capture meaningful spatial expression patterns beyond variance-based (PCA) or expression-only (scGPT) features.

Finally, we assess the biological plausibility of the constructed gene graphs by validating their edges against the STRING protein–protein interaction database on the HGBM dataset (14577 genes). STRING is a widely used resource of experimentally validated and predicted protein–protein functional interactions. Specifically, we select STRING DB v12.0 with Score $>=$ 400. We construct three gene graphs, including (i) Co-expression only, where edges represent co-expression; (ii) GenePT only, where edges represent GenePT-predicted interactions; and (iii) SpaEF, which combines GenePT with co-expression. The results are reported in Table 9. Precision is defined as the fraction of inferred edges that are validated by STRING among all inferred edges. The experimental results show that combining GenePT with co-expression yields the highest proportion of STRING-supported edges.

## G. Sensitivity Analyses

In this section, we assess the influence of the values of $k_s$, $k_g$, $k_e$, and $\tau$ used in spot and gene graph construction. The evaluation metrics are the correlation between PCNA gene expression and histology staining intensity on the HGBM dataset, and the clustering ARI and ACC results on the DLPFC dataset. The values of the ARI and ACC metrics are the average results of five time experiments. As shown in Table 10, setting $k_s, k_g = 5$ results in under-connected graphs that fail to capture sufficient relationships, whereas $k_s, k_g = 15$ introduces noisy or spurious edges that obscure meaningful biological relationships. By contrast, the intermediate choice of $k_s, k_g = 10$ (the adopted setting) achieves the best performance. As shown in Table 11, $k_e = 50$ yields the best result. Finally, we assess the influence of varying values of threshold $\tau$, which determines whether a gene interaction provided by the LR predictor is accepted during the gene graph construction. As shown in Table 12, when $\tau$ is set to $0.5$ (the adopted setting), SpaEF achieves the best performance.

