# OpenReview forum: "SpaEF: Spatially Resolved Transcriptomics Data Element-Wise Denoising Framework Powered by Large Models"
_ICML.cc/2026/Conference — ICML 2026 regular_

### Official Review · Reviewer_hydm · 2026-03-12

**Soundness:** 3
**Presentation:** 3
**Significance:** 3
**Originality:** 3
**Overall Recommendation:** 4
**Confidence:** 4

**Summary:**

This paper presents a novel spatial transcriptomics denoising framework named SpaEF, which leverages pre-trained large models (OmicLIP and GenePT) to construct spot and gene graphs, and introduces an element-wise graph autoencoder (EGAE) for fine-grained denoising. The topic is timely and important, given the high noise levels in spatially resolved transcriptomics data and the growing interest in foundation models for biomedical applications. The authors make several notable efforts, including the integration of multimodal pre-trained LMs and the design of element-wise weighting for graph fusion.

**Compliance With Llm Reviewing Policy:**

Affirmed.

**Key Questions For Authors:**

1．Could the authors provide quantitative evidence (e.g., ARI for clustering, detection of spatially variable genes) demonstrating how SpaEF's denoised data improves downstream biological analyses compared to raw data or other denoising methods?

2．Would the authors consider extending the sensitivity analysis to additional datasets (e.g., DLPFC, HDHBC) or exploring the impact of architectural choices (e.g., number of GAT layers, embedding dimensions) to better understand parameter generalizability and model optimization?

**Limitations:**

yes

**Strengths And Weaknesses:**

Strengths:

1. Innovative Integration of Pre-trained LMs: The use of OmicLIP and GenePT to extract semantic spot and gene features is a novel and promising direction. This approach reduces reliance on noisy histological images and captures nonlinear gene relationships beyond co-expression.
2. Comprehensive Evaluation: The authors conduct extensive experiments on four real-world datasets (HOCWT, HGBM, HDHBC, DLPFC) with diverse characteristics (e.g., high sparsity, multi-modal validation). The use of RMSE, PCC, SCC, and spatial visualizations strengthens the validity of the claims.
3. Robustness and Ablation Studies: The inclusion of edge perturbation tests, modularity analysis, and comparisons with alternative feature extractors (scGPT, PCA) demonstrates the robustness and structural validity of the constructed graphs.
4. Element-wise Denoising: The proposed EGAE module with element-wise weighting is a clear improvement over matrix-wise fusion methods, preserving heterogeneity and improving denoising granularity.

Weaknesses:
1. The paper emphasizes that SpaEF preserves biological relationships while reducing spurious signals. To further support this claim and demonstrate practical utility, it would be valuable to quantitatively evaluate how the denoised data benefits downstream biological analyses, such as clustering, spatial domain identification, or differential expression. Including metrics like adjusted Rand index (ARI) for clustering or detection of spatially variable genes could help illustrate the biological relevance of the method.
2. The paper includes a sensitivity analysis for $k_s,k_g,k_e$, and $\tau$ on the HGBM dataset (Appendix F), which helps understand the impact of these key parameters. Extending this analysis to additional datasets—such as DLPFC or HDHBC, which differ in tissue type, sparsity, and platform—could offer a broader perspective on parameter stability and generalizability. Alternatively, a practical guideline or heuristic for parameter selection in new datasets would be a helpful addition for users. Moreover, while the current analysis focuses on graph construction parameters, exploring the influence of architectural choices (e.g., number of GAT layers, embedding dimensions, attention heads) could further inform model design and optimization.
3. The source code contains only basic code without detailed documentation or example data. To facilitate reproducibility and wider adoption, it would be helpful to provide a well-documented repository with clear instructions, example datasets, and scripts that reproduce the main results.
4. The bar plots in Figures 4 and 5 are informative. Adding error bars (e.g., standard deviation across multiple runs) would improve interpretability and give a clearer sense of result stability, particularly for methods that may exhibit higher variability.

---

> ### Author Rebuttal · Authors · 2026-03-31
>
> Thank you for your appraisal and constructive comments! We are more than happy to discuss any further concerns or questions that you may have.
>
> ---
> **R4.1 (Downstream Analyses):**
>
> We agree with you that evaluating SpaEF on practical downstream applications is crucial. In this regard, our submission had conducted experiments on extensive downstream applications, e.g., gene-protein consistency task (Sections 4.2&4.4), gene-gene relationship recovery task (Section 4.2), and spatial expression pattern recovery task (Sections 4.2&4.3). These results validate SpaEF's effectiveness on diverse downstream applications.
>
> Following your valuable suggestion, we further conduct non-spatial and spatial experiments on DLPFC. Results are quantified with two metrics: Adjusted Rand Index (ARI) and accuracy (ACC). Due to the limited time available during the rebuttal period, we only compare SpaEF with several methods that have demonstrated better performance on the spatial expression pattern recovery task. The results show that SpaEF improves domain identification and clustering quality compared to baselines, indicating better preservation of domain-level or cluster-level structures.
>
> The results in our submission, along with new ones shown above, demonstrate that SpaEF can preserve or enhance the biological characteristics of SRT data. We will include the newly added results in our manuscript.
>
> |Methods|Clustering.ARI|Clustering.ACC|Domain.ARI|Domain.ACC|
> |-|-|-|-|-|
> |raw|0.040|0.308|0.262|0.469|
> |Sprod|0.270|0.446|0.274|0.457|
> |STAGATE|0.282|0.467|0.338|0.537|
> |DUSTED|0.271|0.488|0.325|0.511|
> |DeepGFT|0.295|0.465|0.306|0.480|
> |**SpaEF**|**0.297**|**0.501**|**0.353**|**0.544**|
>
>
> ---
> **R4.2 (Sensitivity Analysis of Hyperparameters):**
>
> We agree with you that evaluating the sensitivity of hyperparameters is important for assessing robustness and generalizability. Accordingly, Appendix F had presented sensitivity analyses of $k_s$, $k_g$, $k_e$, and $\tau$ on HGBM. Following your valuable suggestion, we further conduct experiments on DLPFC, which differs from HGBM in tissue characteristics and spatial expression patterns. As shown in the table below, these hyperparameters are robust.
>
> Based on both the original and newly added results, we deem SpaEF is robust to moderate variations in these hyperparameters. Therefore, dataset-specific tuning or adaptive hyperparameter selection is not necessary in SpaEF.
>
> |$k_s$|$k_g$|Clustering.ARI|Clustering.ACC|
> |-|-|-|-|
> |5|5|0.182|0.417|
> |5|10|0.192|0.417|
> |5|15|0.187|0.424|
> |10|5|0.240|0.453|
> |**10**|**10**|**0.297**|**0.501**|
> |10|15|0.239|0.447|
> |15|5|0.251|0.469|
> |15|10|0.230|0.443|
> |15|15|0.261|0.467|
>
> |$k_e$|Clustering.ARI|Clustering.ACC|
> |-|-|-|
> |15|0.253|0.478|
> |25|0.234|0.465|
> |**50**|**0.297**|**0.501**|
> |65|0.268|0.480|
> |75|0.240|0.465|
>
> |$\tau$|Clustering.ARI|Clustering.ACC|
> |-|-|-|
> |0.4|0.251|0.462|
> |**0.5**|**0.297**|**0.501**|
> |0.6|0.275|0.485|
>
> Regarding architectural hyperparameters (e.g., number of GAT layers, embedding dimensions, attention heads), we follow the configurations adopted in prior arts [4,7] to ensure fair comparison. Due to the limited time during the rebuttal period, we are currently unable to provide a comprehensive ablation study on architectural choices. We will include such analyses in the revised manuscript to further investigate their impact on model performance.
>
> [4] DUSTED: Dual-Attention Enhanced Spatial Transcriptomics Denoiser, AAAI, 2025.
>
> [7] DeepGFT: identifying spatial domains in spatial transcriptomics of complex and 3D tissue using deep learning and graph Fourier transform, Genome Biol., 2025
>
> ---
> **R4.3 (Code Documentation):**
>
> Following your helpful suggestion, we have updated our anonymous source code link to include a detailed README with step-by-step instructions, example datasets, and scripts to reproduce the main results.
>
> ---
> **R4.4 (Std of Results):**
>
> Following your valuable suggestion, we further evaluate the stability of SpaEF by conducting experiments with five different random seeds. We additionally select the two strongest baselines based on their performance on the PCNA.gene–PCNA.protein (PCC/SCC) consistency recovery task, and report their mean and standard deviation for comparison with SpaEF. As shown in the table below, SpaEF exhibits consistently stable performance across runs, further confirming its robustness.
>
> |Method|PCNA.gene–PCNA.protein (PCC/SCC)|CD44.gene–MYC.gene (PCC/SCC)|
> |-|-|-|
> |STAGATE|0.503 ± 0.020 / 0.563 ± 0.022|0.523 ± 0.133 / 0.579 ± 0.129|
> |DUSTED|0.583 ± 0.008 / 0.649 ± 0.009|**0.807 ± 0.031 / 0.855 ± 0.024**|
> |**SpaEF**|**0.602 ± 0.005 / 0.666 ± 0.005**|0.626 ± 0.021 / 0.742 ± 0.023|

---

> > ### Author Rebuttal · Reviewer_hydm · 2026-04-03
> >
> > The reviewer thanks the authors for their responses.

---

> > > ### Author Response · Authors · 2026-04-03
> > >
> > > We are rather delighted to know that our clarifications and additional experiments have adequately addressed your concerns, and we sincerely appreciate your positive assessment of both the innovation of SpaEF and the comprehensiveness of our evaluation.

---

### Official Review · Reviewer_UxoZ · 2026-03-12

**Soundness:** 3
**Presentation:** 3
**Significance:** 3
**Originality:** 3
**Overall Recommendation:** 5
**Confidence:** 4

**Summary:**

Spatial transcriptomics (more broadly, spatial -omics) is an impactful and relatively new high-throughput technology in molecular biology, enabled by recent technology advances (from companies such as 10x Genomics). Intact tissue sections and slices are probed for the spatial mRNA expression of a broad range of genes, with a variety of methods such as through arrays with barcoded oligonucleotide probes being applied to the tissue slices, through imaging, or mRNA capture and nucleotide sequencing.

All technologies to date provide information on gene expression with spatial resolution, but at the same time suffer from a wide array of systematic noise sources such as preanalytical variation (including tissue handling and microdisection), natural gene fluctuations, as well as the sources of error introduced in each step of the technology's workflow. However, high spatial resolution results in gene expression values in nearby locations being correlated. Denoising methods attempt to leverage such correlations without blunting the fine-grain substructure of the tissue, which is revealed after spatial transcriptomic analysis.

The authors present a new spatial transcriptome denoising method, SpaEF, which introduces novel methodological features, and importantly, is experimentally validated against state-of-the-art methods (except for the scGPT family of methods, for which the authors could not get properly running code), on four real datasets. The method is reported to achieve best performance. There is also an informative ablation study showing that their introduced methodological features contribute to the performance.

**Compliance With Llm Reviewing Policy:**

Affirmed.

**Final Justification:**

I think this is a sound ML method, of reasonable originality, with some novel methods introduced. Importantly, it is tested on real biological data and demonstrated to perform at state-of-the-art level. The biological problem is current and important. So, overall, this is a good contribution.

**Key Questions For Authors:**

1. It would be great if comparison with scGPT is included. The PI, Bo Wang, is usually highly responsive (from the reviewer's personal experience) and would likely help rapidly installing the code.

2. For exposition purposes, please include a brief description of each of the tested datasets.

**Limitations:**

Yes

**Strengths And Weaknesses:**

A key strength of the method is that it addresses an important practical problem in computational biology, spatial transcriptomics, is tested on four diverse real datasets against state-of-the-art, and demonstrates best performance. This ensures that the work will have meaningful impact.

The authors introduce state-of-the-art methodology including leveraging pre-trained models and integrating them using graph attention networks.

In terms of methodological novelty, the authors claim that theirs is the first spatial transcriptomic denoising method that uses pretrained large models to construct spot and gene graphs. It is not easy to evaluate how narrow or broad this claim should be. In particular, scGPT-spatial (a method that the authors don't test against), is based on continual pretraining of a model based on the foundation model scGPT. The authors claim that "SpaEF is the first SRT data denoising work that leverages pre-trained LMs for constructing spot and gene graphs", and depending on how narrowly this is interpreted, it could be true, but SpaEF does not seem to be the first spatial transcriptomics method that uses pretrained foundation models.

---

> ### Author Rebuttal · Authors · 2026-03-31
>
> Thank you for your appraisal and constructive suggestions regarding our research! We are more than happy to discuss any further concerns or questions that you may have.
>
> ---
> **R3.1 (Difference between SpaEF and scGPT-spatial):**
>
> We would like to clarify that the key distinction between SpaEF and other methods lies in how large models (LMs) are utilized. Specifically, SpaE leverages pretrained LMs as feature generators to provide external biological and semantic priors, where other methods (e.g., scGPT-spatial) employ LMs as representation learners, learning embeddings for input data, which may generate low-quality embeddings in zero-shot [5], while fine-tuning consumes much time and resources. Thus, although these methods employ pretrained LMs, their usage paradigm differs fundamentally from that of SpaEF. Due to the limited time during the first-stage rebuttal and the lack of a complete and reproducible implementation of scGPT-spatial (see R3.2), we are currently unable to include this baseline. If a runnable implementation becomes available in the second-stage rebuttal, we will incorporate the comparison to further strengthen the evaluation.
>
> In addition, to effectively incorporate pretrained LM knowledge into SRT denoising without introducing domain mismatch or disrupting data-specific structures, SpaEF introduces dedicated modules rather than directly applying LMs. Specifically, it: (1) integrates LM-derived spot features with spatial coordinates via the proposed AGAE (see Section 3.1), thereby mitigating semantic–spatial misalignment; (2) refines GenePT-inferred gene relationships using co-expression-based pruning (see Section 3.2), ensuring consistency with dataset-specific signals; (3) performs element-wise addition (see Section 3.3) to enable effective message passing between spot and gene graphs.
>
> Based on these contributions, we believe that our claim is appropriately scoped. We will revise the manuscript to clarify the distinction between SpaEF and other methods following your valuable comment, thereby highlighting the positioning and contributions of SpaEF.
>
> [5] Zero-shot evaluation reveals limitations of single-cell foundation models, Genome Biol., 2025.
>
> ---
> **R3.2 (Additional Baselines):**
>
> We agree with you that including more baselines would further strengthen the empirical evaluation of SpaEF. Per your valuable suggestion, we have attempted to contact the scGPT-spatial team via email. However, we have not yet received a response. As the currently available implementation of scGPT-spatial is incomplete and lacks essential components for reproduction, we are unable to include it as a baseline at this stage. If the full implementation becomes available during the discussion period, we will promptly incorporate it into our comparisons.
>
> To further ensure a comprehensive evaluation, we additionally include SpotSpace [6] as a baseline. As shown in the following table, SpaEF consistently demonstrates superior performance on the gene-protein consistency task and gene-gene relationship recovery task over SpotSpace on HGBM and HDHBC datasets. These additional results further demonstrate the effectiveness of SpaEF and will be included in the revised manuscript to strengthen the empirical evaluation.
> |Method|PCNA.gene–PCNA.protein (PCC/SCC)|CD44.gene–MYC.gene (PCC/SCC)|PTPRC.gene–CD45RO.protein (PCC/SCC)|
> |-|-|-|-|
> |SpotScape|0.503/0.569|0.416/0.581|0.027/0.023|
> |SpaEF|0.611/0.675|0.652/0.768|0.053/0.050|
>
> [6] Global Context-aware Representation Learning for Spatially Resolved Transcriptomics, ICML 2025.
>
> ---
> **R3.3 (Description of Datasets):**
>
> We fully agree with your view that providing clear descriptions of the test datasets is important for better exposition. Due to the space limitations in the main text, we had included detailed descriptions of the masked real-world dataset (HOCWT) and three real-world datasets (HGBM, HDHBC, and DLPFC), along with their preprocessing procedures, in Appendix B.
>
> Following your valuable suggestion, we will further improve the exposition by explicitly referencing the appendix in the main text of the revised manuscript.

---

> > ### Author Rebuttal · Reviewer_UxoZ · 2026-04-02
> >
> > Thank you for your answers to my questions. I have already marked the paper as "accept", and hope it does get accepted.

---

> > > ### Author Response · Authors · 2026-04-03
> > >
> > > We sincerely appreciate the reviewer's time, expertise, and constructive comments, as well as the positive appraisal of our work!
> > >
> > > We also apologize for the delayed response. During this time, we conduct additional comparative experiments with scGPT-spatial on the HGBM and HDHBC datasets. As shown in the table below, the results further demonstrate the effectiveness of our SpaEF. We will include these experiments in the revised manuscript to further strengthen the empirical evaluation!
> > >
> > > |Method|PCNA.gene–PCNA.protein (PCC/SCC)|CD44.gene–MYC.gene (PCC/SCC)|PTPRC.gene–CD45RO.protein (PCC/SCC)|
> > > |-|-|-|-|
> > > |Sprod|0.510/0.573|0.497/0.640|0.014/0.021|
> > > |Smoother|0.369/0.401|0.163/0.160|-0.003/0.009|
> > > |STAGATE|0.532/0.591|0.567/0.620|0.037/0.045|
> > > |DUSTED|0.586/0.653|0.774/0.840|0.018/0.037|
> > > |DeepGFT|0.323/0.387|0.570/0.743|0.041/-0.002|
> > > |scGPT-spatial|0.365/0.222|0.368/0.223|0.027/0.024|
> > > |**SpaEF**|0.611/0.675|0.652/0.768|0.052/0.050|

---

### Official Review · Reviewer_UMtm · 2026-03-13

**Soundness:** 3
**Presentation:** 3
**Significance:** 3
**Originality:** 3
**Overall Recommendation:** 4
**Confidence:** 3

**Summary:**

The article proposes using pre-trained Large Models (LMs) for denoising spatially resolved transcriptomic data. Specifically, the framework uses OmniCLIP for spot feature extraction and GenePT for non-linear relationships among genes. In addition, the article proposes two encoders for integrating extracted spot features with spatial coordinates and fusing the spot and gene graphs. Several benchmark datasets have been used to compare the performance of the proposed framework with existing frameworks.

**Compliance With Llm Reviewing Policy:**

Affirmed.

**Key Questions For Authors:**

1. Could you provide more detail on how AGAE and EGAE loss functions have been optimized?
2. How would you detect if there is any mismatch between the LMs output and biological pathways?
3. Is overdispersion of gene expression a concern, and how is it accommodated in EGAE?

**Limitations:**

Limitations have not been described.

**Strengths And Weaknesses:**

Strengths: The problem is well-motivated and the method is well-explained. Comparison with other methods is extensive.

Weaknesses: The method has been demonstrated mostly empirically. The training of the AGAE and EGAE have not been fully described in the appendix. Potential mismatch arising between neural embedding and biological mechanism have not been discussed.

---

> ### Author Rebuttal · Authors · 2026-03-31
>
> Thank you for your appraisal, as well as your insightful and constructive comments! If anything remains unclear, we would be glad to offer further clarification.
>
> ---
> **R2.1 (Training Process of AGAE and EGAE):**
>
> We agree that a more detailed description of the training process would improve clarity and reproducibility. In our submission, we had provided the loss functions of AGAE and EGAE in Appendix A, along with the corresponding hyperparameters in Table 4. Following your suggestion, we are happy to further elaborate on the training procedure. Due to the character limit, we briefly outline the key steps below and will include the full details in the revised manuscript to improve clarity and reproducibility.
>
> The training objective of AGAE is to minimize a weighted binary cross-entropy loss (Eq. (20)) between the reconstructed adjacency and the adjacency of the spatial graph using the Adam optimizer. Gradients are backpropagated to update the learnable weights to minimize this loss, thereby enabling AGAE to learn spot embeddings. Subsequently, EGAE is optimized using the Adam optimizer with a Zero-Inflated Negative Binomial (ZINB) loss (Eq. (23)). The gradients derived from ZINB loss are backpropagated to update the parameters of the EGAE.
>
> If any aspect of the training process remains unclear, we would be happy to provide further clarification.
>
> ---
> **R2.2 (Mismatch between LMs' Embeddings and Biological Pathways):**
>
> We sincerely thank you for this insightful question. Because LM-derived embeddings do not inherently encode explicit biological pathway information, the accuracy of edges in the graphs constructed by LM-derived embeddings is typically evaluated to assess the alignment between these embeddings and actual biological pathways [2]. Meanwhile, effectively matching LM-derived embeddings with biological pathways, manifested as accurately capturing spot–spot and gene–gene relationships in graphs, remains a critical challenge.
>
> To obtain biologically meaningful edges, we propose AGAE, which grounds the LM-derived semantic embeddings within the spatial coordinates of the spots (see Lines 148-152). Moreover, our GGC module uses co-expression constraints to tailor the global biological priors extracted by GenePT to specific tissue inputs (see Lines 229-233).
>
> The results in Appendix E demonstrate that the edges of our spot graph and gene graph are biologically meaningful, indicating that our design effectively aligns LM-derived embeddings with biological pathways. We will include this analysis in our manuscript to further support this claim.
>
> [2] Integrating and formatting biomedical data as pre-calculated knowledge graph embeddings in the Bioteque, Nat. Commun., 2022.
>
> ---
> **R2.3 (Overdispersion of Gene Expression):**
>
> The overdispersion of the expression matrix is a critical challenge in SRT data. To better address this challenge, we follow the data preprocessing in [3] and the loss function in [4].
>
> In the data preprocessing stage, as detailed in Appendix B, we first filter out low-quality genes and spots to mitigate the initial overdispersion. We then apply Counts Per Million normalization and a log1p transformation to resolve the long-tail distributions and variance instability caused by overdispersion.
>
> In addition, to train EGAE, we adopt the Zero-Inflated Negative Binomial (ZINB) loss function (see Appendix A). Specifically, the Negative Binomial (NB) component of this loss is rigorously formulated for overdispersed count data, enabling EGAE to fundamentally and mathematically adapt to the overdispersed nature of gene expression.
>
> As shown in the results on HOCWT and DLPFC datasets (both known to exhibit overdispersion), our SpaEF outperforms other baselines, demonstrating its effectiveness in handling overdispersed data.
>
> [3] Sprod for de-noising spatially resolved transcriptomics data based on position and image information, Nat. Methods, 2022.
>
> [4] DUSTED: Dual-Attention Enhanced Spatial Transcriptomics Denoiser, AAAI, 2025.
>
> ---
> **R2.4 (Limitations):**
>
> We agree that explicitly discussing the limitations of SpaEF is valuable for the community. We acknowledge that, although SpaEF achieves the best denoising results on the HDHBC dataset compared to other methods, the absolute performance metrics remain poor, indicating that SpaEF still cannot effectively denoise extremely sparse datasets. We will include a dedicated discussion of these limitations in the revised manuscript and further outline potential directions for addressing this issue.

---

> > ### Author Rebuttal · Reviewer_UMtm · 2026-04-05
> >
> > The authors have adequately responded to my questions.

---

> > > ### Author Response · Authors · 2026-04-05
> > >
> > > We are delighted to know that our clarifications have adequately addressed your concerns. We also sincerely appreciate the reviewer’s positive assessment. We are encouraged that the motivation of our work is recognized as well-founded, the proposed method as clearly presented, and the experimental comparisons as comprehensive.

---

### Official Review · Reviewer_XMDg · 2026-03-13

**Soundness:** 3
**Presentation:** 4
**Significance:** 2
**Originality:** 1
**Overall Recommendation:** 3
**Confidence:** 5

**Summary:**

The authors argue that denoising for spatial transcriptomics (SRT) is limited by prior methods that fail to jointly leverage spot-level information and gene–gene relationships, and in some cases introduce bias during modeling. To address these limitations, the proposed method, SpaEF, constructs both a spot graph and a gene graph to capture both relationships. It extracts feature representations from two large pre-trained models: OmiCLIP for spot-level features and GenePT for gene-level representations. Subsequently, an EGAE module performs denoising on each graph separately and integrates the results to achieve improved denoising performance. Through extensive experiments, the authors demonstrate that their approach achieves superior performance compared with existing methods.

**Compliance With Llm Reviewing Policy:**

Affirmed.

**Final Justification:**

While I still have concerns about the novelty of this work, I agree with its practicality following the additional experiments conducted during the rebuttal period.

**Key Questions For Authors:**

1. The main weakness of this paper is that the authors focus primarily on recovering masked expressions, evaluated using RMSD and correlation metrics. However, a more important aspect is whether the method can preserve or enhance the biological characteristics of SRT data. To better demonstrate this, the authors should evaluate the method on representative downstream tasks, such as spatial domain identification or trajectory inference.

2. In addition, the paper would benefit from including case studies. For example, demonstrating that the method helps recover biologically meaningful pathways or improves pathway-level analysis would provide stronger evidence of its practical utility.

3. The proposed method naively fuses the denoising results from the two branches. However, there are various alternative design choices that could be explored. For example, the authors could consider attention-based fusion or a sequential denoising strategy, such as first denoising the SpotGraph and then refining the results using the GeneGraph.

4. The authors also do not include some recent baseline methods. For instance, methods such as SpotScape [1], which have recently reported strong performance on denoising tasks, should be included for a more comprehensive comparison.

[1] Global Context-aware Representation Learning for Spatially Resolved Transcriptomics, ICML 2026

**Limitations:**

As discussed above, this paper lacks sufficient evidence demonstrating its impact on downstream tasks. The authors should include evaluations on practical downstream analyses after denoising, in order to show whether the proposed method improves performance in real biological applications.

**Strengths And Weaknesses:**

* Soundness: The proposed method is well aligned with the stated motivation. Previous works sometimes rely on histological images, and many methods fail to adequately capture relationships between genes. The proposed framework attempts to address these limitations by incorporating both spot-level and gene-level information.

* Presentation: The paper is well organized and generally easy to follow. The methodology and overall pipeline are presented clearly, which helps readers understand the proposed framework.

* Significance: The paper addresses an important problem in spatial transcriptomics, which is relevant for biological discovery. However, the evaluation mainly focuses on computational recovery of masked values. The study does not sufficiently demonstrate whether the method leads to improved biological insights or practical downstream applications, which limits the overall impact.

* Originality: In my view, the level of novelty is limited. Although the paper introduces large models for SRT denoising, it primarily utilizes already pre-trained models and combines their representations in a relatively straightforward manner. As a result, the methodological contribution appears incremental rather than fundamentally new.

---

> ### Author Rebuttal · Authors · 2026-03-31
>
> Thanks for your comments. Please let us know if you have any further questions.
>
> ---
> **R1.1 (Downstream Applications):**
>
> We agree with your view that evaluating SpaEF on practical downstream applications is crucial. In this regard, our submission had conducted experiments on extensive downstream applications, e.g., gene-protein consistency task (**Sections 4.2&4.4**), gene-gene relationship recovery task (**Section 4.2**), and spatial expression pattern recovery task (**Sections 4.2&4.3**), all of which directly assess biologically meaningful characteristics rather than merely recovering masked expressions. These results validate SpaEF's effectiveness on diverse downstream applications.
>
> Nonetheless, per your comment, we have conducted spatial domain identification and clustering experiments on DLPFC. Due to space limitations, please see **R4.1 to Reviewer hydm** for more details.
>
> In summary, **both the results in our submission and the newly added ones consistently validate that SpaEF can preserve or enhance the biological characteristics of SRT data.**
>
> ---
> **R1.2 (Novelty):**
>
> **We respectfully disagree with your view regarding the novelty of SpaEF.** First, as explicitly clarified in our submission (e.g., **Lines 73–83**), SpaEF goes beyond directly leveraging OmiCLIP and GenePT. A key challenge is that these LMs are not directly compatible with SRT denoising, because OmiCLIP-derived embeddings need to be integrated with the spatial context (see **Lines 149-151**), and GenePT-provided gene interaction priors lack the input-specific nature of SRT data (see **Lines 229–233**).
>
> To address this challenge, we propose GAT-based AGAE to integrate OmiCLIP-derived features with spatial coordinates and use co-expression constraints to tailor the global biological priors extracted by GenePT to specific tissue inputs. Moreover, to enable effective message passing between spot and gene graphs, we also propose an element-wise addition mechanism (see **Section 3.3**). Finally, extensive results validate that each module's contribution (see **Table 3**).
>
> Thus, we believe that **our contributions are not overstated and SpaEF provides sufficient innovation, meriting consideration for acceptance at the prestigious ICML.**
>
> ---
> **R1.3 (Case Studies):**
>
> We agree that SpaEF would benefit from including case studies, and such case studies are already included in our submission (see R1.1).
>
> In particular, pathway-level analysis is fundamentally characterized by coordinated interactions among functionally related genes [1]. In this regard, our experiment on the HGBM dataset (see **Lines 370-382**) had analyzed the restored correlation between CD44 and MYC genes, thereby providing evidence that SpaEF is capable of recovering biologically meaningful interactions at the pathway level.
>
> Therefore, we deem **our submission had demonstrated SpaEF's effectiveness and practical utility**.
>
> [1] Ten years of pathway analysis: current approaches and outstanding challenges. PLoS Comput. Biol., 2012.
>
> ---
> **R1.4 (Fusion Strategy):**
>
> As discussed in **Lines 71-78**, due to the element-level heterogeneity inherent in SRT data, matrix-wise weighting may limit the model’s ability to adaptively denoise individual elements. To address this limitation, we propose a straightforward yet effective Element-Wise Addition (EWA) strategy. In addition, as shown in **Table 3,** we had explored the other fusion strategies, e.g., the cross-attention strategy, validating the effectiveness of EWA.
>
> Per your comment, we further conduct ablation studies on the gene-protein consistency task in the HGBM dataset with two sequential denoising methods you mentioned, namely S-A (first spot graph then gene graph) and S-B (first gene graph then spot graph). As shown in the table below, EWA consistently outperforms others.
>
> In summary, experiments in our submission, along with new results shown below, validate that **EWA is not a naive design, but a well-motivated and effective strategy that explicitly accounts for element-level heterogeneity in SRT data**.
>
> |Fusion Strategy|PCC|SCC|
> |-|-|-|
> |cross-attention|0.555|0.622|
> |S-A|0.537 |0.590|
> | S-B|0.528|0.603|
> | EWA|0.611|0.675|
>
> ---
> **R1.5 (Additional Baseline):**
>
> To assess the performance of SpaEF, our submission had included SOTA baselines, e.g., DUSTED (AAAI'25) and DeepGFT (GenomeBiol.’25). Extensive results demonstrate the advantages of our SpaEF.
>
> Per your comment, we have compared SpaEF with SpotScape. As shown in the table below, the experiments on HGBM and HDHBC datasets validate that SpaEF outperforms SpotScape with the PCC/SCC evaluation.
>
> Therefore, we deem **SpaEF had been comprehensively evaluated and provided a meaningful performance gain over SOTA baselines across all datasets.**
>
> |Method|PCNA.gene–PCNA.protein (PCC/SCC)|CD44.gene–MYC.gene (PCC/SCC)|PTPRC.gene–CD45RO.protein (PCC/SCC)|
> |-|-|-|-|
> |SpotScape| 0.503/0.569| 0.416/0.581|0.027/0.023|
> |SpaEF|0.611/0.675|0.652/0.768|0.053/0.050|

---

> > ### Author Rebuttal · Reviewer_XMDg · 2026-04-06
> >
> > The authors provide additional empirical results, such as spatial domain identification, and report comparisons with different fusion strategies and additional baselines. These results are sufficient to address my previous concerns. I will raise my score.

---

> > > ### Author Response · Authors · 2026-04-06
> > >
> > > We are delighted to know that our clarifications have adequately addressed your concerns. We sincerely thank you for raising your score and for your support of our work!
> > >
> > > We would like to take this opportunity to highlight the contributions of our work. While certain studies use LM as a denoiser to directly generate denoised outputs, such approaches may suffer from limited output quality and introduce additional computational overhead (see **R3.1** to Reviewer UxoZ). In contrast, to the best of our knowledge, **SpaEF is the first to introduce a paradigm for integrating LMs into SRT data denoising by leveraging them as encoders to inject biological prior knowledge.**
> > >
> > > However, this paradigm introduces non-trivial challenges. Specifically, **LM-derived representations are not directly compatible with SRT denoising**: OmiCLIP-derived embeddings lack spatial context, while GenePT-provided gene interaction priors do not capture the input-specific nature of SRT data. To address these challenges, we propose **AGAE** to complement the OmiCLIP embedding with spatial information, and **GGC** to adapt the global biological priors from GenePT to specific tissue inputs via co-expression constraints. Furthermore, to handle the inherent element-level heterogeneity in SRT data, we design the **EWA** strategy to enable fine-grained, element-wise denoising (see **R1.4**). While certain modules may be conceptually simple, their design is carefully tailored to the characteristics of the SRT denoising task, making them both effective and complementary. Ablation studies (see **Table 3**) demonstrate that each proposed module contributes effectively and robustly to SpaEF, and, along with graph analysis (see **Appendix E**), shows that directly applying LMs fails to achieve high-level performance.
> > >
> > > Therefore, we believe that **SpaEF represents a genuinely novel paradigm rather than merely an SRT denoising model that adopts pre-trained LMs.**

---

### Decision · Program_Chairs · 2026-04-30

**Decision:**

Accept (regular)

**Comment:**

This paper drew mixed initial scores, with the main concerns being biological utility beyond masked-value recovery, justification of the fusion design, missing baselines, and the scope of the novelty claim; however, after reading the reviews, rebuttal, and discussion, I find that the authors addressed most of these concerns through added downstream evaluation, additional baseline and fusion comparisons, clarifications of training/details, and further robustness analyses, and several reviewers explicitly indicated that their concerns were resolved. Overall, I view the paper as technically sound, relevant, and empirically strong on an important application problem, though I agree that the novelty is somewhat moderate and that the “first” claim should be scoped more carefully relative to related foundation-model-based methods. On balance, I believe the paper is above the bar for acceptance.